# Geometric Analysis of Matrix Sensing over Graphs

**Haixiang Zhang**
Department of Mathematics
University of California, Berkeley
Berkeley, CA 94720
haixiang_zhang@berkeley.edu

**Ying Chen**
Department of IEOR
University of California, Berkeley
Berkeley, CA 94720
ying-chen@berkeley.edu

**Javad Lavaei**
Department of IEOR
University of California, Berkeley
Berkeley, CA 94720
lavaei@berkeley.edu

## Abstract

In this work, we consider the problem of matrix sensing over graphs (MSoG). As a general case of matrix completion and matrix sensing problems, the MSoG problem has not been analyzed in the literature and the existing results cannot be directly applied to the MSoG problem. This work provides the first theoretical results on the optimization landscape of the MSoG problem. More specifically, we propose a new condition, named the $\Omega$-RIP condition, to characterize the optimization complexity of the problem. In addition, with an improved regularizer of the incoherence, we prove that the strict saddle property holds for the MSoG problem with high probability under the incoherence condition and the $\Omega$-RIP condition, which guarantees the polynomial-time global convergence of saddle-avoiding methods. Compared with state-of-the-art results, the bounds in this work are tight up to a constant. Besides the theoretical guarantees, we numerically illustrate the close relation between the $\Omega$-RIP condition and the optimization complexity.

## 1 Introduction

In a wide range of problems in the fields of machine learning, signal processing and power systems, an unknown low-rank matrix parameter should be estimated from a few measurements of the matrix. To be more specific, given some measurements of the unknown symmetric and positive semi-definite (PSD) matrix $M^* \in \mathbb{R}^{n \times n}$ of rank $r \ll n$, the *low-rank matrix optimization* problem can be formulated as

$$(1) \qquad \min_{M \in \mathbb{R}^{n \times n}} f(M; M^*) \quad \text{s.t.} \quad M \succeq 0, \quad \text{rank}(M) \leq r,$$

where $f(\cdot; M^*)$ is a loss function that penalizes the mismatch between $M$ and $M^*$. The goal is to recover the matrix $M^*$ via finding a global minimizer of problem (1). Applications of this problem include matrix sensing [31, 40, 38], matrix completion [10, 11, 17], phase retrieval [8, 33, 14], and power systems [39, 24]; see the review papers [13, 15] for more applications. Early attempts to deal with the nonconvex rank constraint of the problem focused on solving a convex relaxation of (1); see [10, 31, 11, 7]. However, the convex relaxation approach usually updates the matrix variable via the Singular Value Decomposition (SVD) in each iteration. This will lead to an $O(n^3)$ computational complexity in each iteration and an $O(n^2)$ space complexity, which are prohibitively

37th Conference on Neural Information Processing Systems (NeurIPS 2023).

high for large-scale problems; see the numerical comparison in [41]. Similar issues are observed for algorithms based on the Singular Value Projection [20] and Riemannian optimization algorithms [35, 36, 19, 1, 27].

To improve the computation and memory efficiency, the Burer-Monteiro factorization approach was proposed in [6], which is based on the fact that the mapping $U \mapsto UU^T$ is surjective onto the manifold of PSD matrices of rank at most $r$, where $U \in \mathbb{R}^{n \times r}$. More concretely, problem (1) is equivalent to

$$\min_{U \in \mathbb{R}^{n \times r}} f(UU^T; M^*). \tag{2}$$

Due to the non-convexity of the mapping $U \mapsto UU^T$, problem (2) is an unconstrained non-convex problem and may have spurious second-order critical points (i.e., second-order critical points that do not correspond to the ground truth matrix $M^*$). In general, saddle-avoiding local search methods are only able to find $\epsilon$-approximate second-order critical points[1]. As a result, local search methods with a random initialization will likely be stuck at spurious second-order critical points and unable to converge to the ground truth solution. However, in a variety of real-world applications, simple algorithms such as perturbed gradient descent methods and alternating minimization methods have achieved empirical success on problem (2). Recently, substantial progress has been made on the theoretical explanation of the benign behavior of these algorithms. For example, the alternating minimization algorithm was studied in [21, 28, 29]. The (stochastic) gradient descent algorithm, which is in general easier to implement than the alternating minimization algorithm, was analyzed in [8, 34, 37, 14, 13]. Moreover, the gradient descent algorithm is proved to have the implicit regularization phenomenon in the over-parameterization case [26, 16, 32].

Besides the algorithmic analysis, a large amount of literature [17, 33, 42, 38] focused on the geometric analysis of the landscape of problem (2), which usually depends on the *strict-saddle property* [33]. Intuitively, the strict-saddle property states that at any feasible point of problem (2), at least one of the three properties will hold: (i) the point is close to a global solution; (ii) the norm of the gradient is large; (iii) the Hessian matrix has a negative eigenvalue. In the later two cases, saddle-escaping algorithms [12, 23, 2] are able to find a descent direction and thus, these algorithms will converge globally in polynomial time. The formal definition of the strict-saddle property is provided in Section 3.1. In the following, we review the state-of-the-art conditions for two special classes of problem (2) that guarantee the strict-saddle property; see the survey [15] for other problem classes.

## 1.1 Matrix sensing and the Restricted Isometry Property (RIP) condition

In the matrix sensing problem, the information of the ground truth matrix $M^*$ is gathered via the measurement operator $\mathcal{A}$. The loss function $f$ is usually chosen to be the negative log-likelihood function. For example, in the classic linear matrix sensing problem, the operator $\mathcal{A}$ is a linear operator defined as

$$\mathcal{A}(M) := [\langle A_1, M \rangle, \ldots, \langle A_m, M \rangle], \quad \forall M \in \mathbb{R}^{n \times n}, \tag{3}$$

where $m$ is the number of measurements and $A_i \in \mathbb{R}^{n \times n}$ contains independently identically distributed (i.i.d) Gaussian random entries and $A_i$'s are independent of each other. If the measurement noise is Gaussian, the maximum likelihood estimation is equivalent to the following minimization problem

$$\min_{U \in \mathbb{R}^{n \times r}} \|\mathcal{A}(UU^T) - \mathcal{A}(M^*)\|_F^2. \tag{4}$$

More examples of the (non-linear) matrix sensing problem are discussed in [42]. One of the most important conditions that guarantee the benign landscapes is the RIP condition:

**Definition 1** (RIP Condition[31, 42]). *Given natural numbers $r$ and $s$, the function $f(\cdot; M^*)$ is said to satisfy the **Restricted Isometry Property** (RIP) of rank $(2r, 2s)$ for a constant $\delta \in [0, 1)$, denoted as $\delta$-RIP$_{2r,2s}$, if*

$$(1 - \delta)\|K\|_F^2 \leq \left[\nabla^2 f(M; M^*)\right](K, K) \leq (1 + \delta)\|K\|_F^2 \tag{5}$$

*holds for all matrices $M, K \in \mathbb{R}^{n \times n}$ such that $\mathrm{rank}(M) \leq 2r$ and $\mathrm{rank}(K) \leq 2s$, where $\left[\nabla^2 f(M; M^*)\right](K, K)$ is the curvature of the Hessian at point $M$ along direction $K$.*

---

[1]A point $x_0$ is called an $\epsilon$-approximate second-order critical point to the optimization problem $\min_x F(x)$ if $\|\nabla F(x_0)\|_F \leq \epsilon$ and $\lambda_{min}[\nabla^2 F(x_0)] \geq -\epsilon$.

For the linear matrix sensing problem (4), it is proved in [9] that the $\delta$-RIP$_{2r,2s}$ condition holds with high probability if $m = \Theta(nr\delta^{-2})$. The RIP condition is also established in many other applications of the matrix sensing problem [42, 4] and is of independent research interest. The constant $\delta$ plays a critical role in characterizing the optimization landscape of problem (2). More specifically, in [5], the authors showed that the strict-saddle property holds for problem (2) if the $\delta$-RIP$_{2r,2r}$ condition holds with $\delta < 1/2$. Counterexamples have been constructed in [40, 38] to illustrate that the strict-saddle property may fail under the $\delta$-RIP$_{2r,2r}$ condition with $\delta \geq 1/2$.

## 1.2 Matrix completion and the incoherence condition

In spite of the strong theoretical results under the RIP condition, there exist a large number of applications that do not satisfy the RIP condition. A well-studied class of problems that does not satisfy the RIP condition is the matrix completion problem. For the matrix completion problem, a subset $\Omega \subset [n] \times [n]$ of entries of $M^*$ are observed and the goal is to recover the low-rank ground truth matrix from the observed entries. For every matrix $M \in \mathbb{R}^{n \times n}$, we denote the projection of $M$ onto $\Omega$ as $M_\Omega$, namely,

$$(M_\Omega)_{ij} = \begin{cases} M_{ij} & \text{if } (i,j) \in \Omega, \\ 0 & \text{otherwise.} \end{cases}$$

Using the $\ell_2$-loss function, the matrix completion problem is defined as

$$(6) \qquad \min_{U \in \mathbb{R}^{n \times r}} \|M_\Omega - M_\Omega^*\|_F^2.$$

The matrix completion problem (6) is a special case of the linear matrix sensing problem (4), where the sample size $m$ is equal to $|\Omega|$ and each measurement matrix $A_i$ contains exactly one nonzero entry. We note that the $\delta$-RIP$_{2r,2r}$ condition does not hold for problem (6) unless we observe all entries of $M^*$, i.e., when $\Omega = [n] \times [n]$. As an alternative to the RIP condition, the incoherence of $M^*$ is useful in characterizing the complexity of problem (6).

**Definition 2** (Incoherence Condition [10]). *Given $\mu \in [1, n]$, the ground truth matrix $M^*$ is said to be $\mu$-incoherent if*

$$(7) \qquad \|e_i^T V^*\|_F \leq \sqrt{\mu r/n}, \quad \forall i \in [n],$$

*where $V^* \Lambda^* (V^*)^T$ is the truncated SVD of $M^*$ and $e_i$ is the $i$-th standard basis of $\mathbb{R}^n$.*

Intuitively, the incoherence of $M^*$ measures the sparsity of the low-rank ground truth matrix. If the incoherence is large, the matrix $M^*$ is highly sparse and it is necessary to measure considerably many entries of $M^*$ to observe nonzero entries. On the other hand, a relatively small incoherence of $M^*$ is able to avoid the extreme case. Except the limited literature on the deterministic matrix completion problem [3, 25, 30], the majority of the matrix completion literature considered the following Bernoulli sampling model:

**Definition 3** (Bernoulli Sampling Model). *Given a sampling rate $p \in (0, 1]$, each index $(i, j) \in [n] \times [n]$ belongs to the set $\Omega$ independently with probability $p$.*

Under the Bernoulli sampling model, the objective function of problem (6) is well-behaved over the set of matrices with a small incoherence. Therefore, a regularizer that penalizes the incoherence of $UU^T$ is included and we instead solve the following regularized matrix completion problem:

$$(8) \qquad \min_{U \in \mathbb{R}^{n \times r}} \frac{1}{p} \left\|(UU^T)_\Omega - M_\Omega^*\right\|_F^2 + \lambda \sum_{i \in [n]} \left(\|e_i^T U\|_F - \alpha\right)_+^4,$$

where $(x)_+ := \max\{x, 0\}$ for all $x \in \mathbb{R}$ and $\alpha, \lambda > 0$ are constants. Intuitively, the coefficient $1/p$ is used to "normalize" the $\ell_2$-norm. We note that there exists an algorithm that can solve problem (6) without the incoherence regularizer [13]. However, the algorithm relies on the spectral initialization, which requires more computational effort than the factorization approach (2). It is proved in [18, 17] that if the sampling rate satisfies

$$(9) \qquad np \geq \Theta[\mu^4 r^6 (\kappa^*)^6 \log n],$$

the problem (8) satisfies the strict-saddle property with high probability, where $\kappa^*$ is the condition number of $M^*$. On the other hand, it is proved in [11] that the information-theoretical lower bound $np \geq \Theta(\mu r \log n)$ is necessary for the exact completion with high probability.

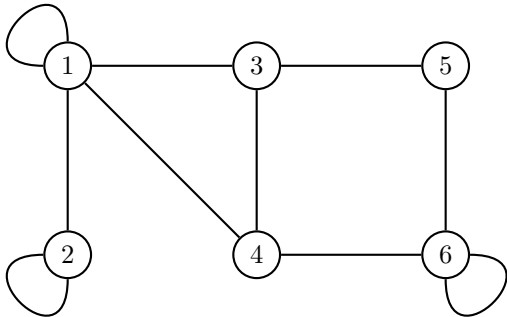

Figure 1: Example of the problem of matrix sensing over graphs. The vertex set is $\mathcal{V} = [6]$ and the edge set is $\mathcal{E} = \{(1,1), (1,2), (1,3), (1,4), (2,2), (3,4), (3,5), (4,6), (5,6), (6,6)\}$.

## 1.3 Motivating example: Matrix sensing over graphs

Although the matrix sensing problem and the matrix completion problem are well-studied in literature, we show that there exist important applications of (2) that belong to a much broader class of problems than the special classes previously studied by the existing works. Hence, the current theoretical results cannot be directly applied to these applications and it remains unknown whether saddle-avoiding algorithms can find the ground truth matrix $M^*$ in polynomial time.

Consider an undirected graph $\mathcal{G} = (\mathcal{V}, \mathcal{E})$, where $\mathcal{V}$ and $\mathcal{E}$ are the set of vertices and the set of edges, respectively. For the notational simplicity, we assume that $\mathcal{V} = [n]$. Then, the edge set $\mathcal{E}$ is a subset of $[n] \times [n]$. The goal of this problem is to recover the ground truth matrix $M^*$ via measurements of its entries $\{M_{ij}^* \mid (i,j) \in \mathcal{E}\}$. Instead of directly observing each entry $M_{ij}^*$, for each node $i \in \mathcal{V}$, we observe a "mixture" (i.e., the output of a function) of the entries in the set

$$\mathcal{E}_i := \{(i,j) \mid j \text{ is incident to } i\}.$$

Denote the loss function for node $i$ as $f_i\left(M_{\mathcal{E}_i}; M_{\mathcal{E}_i}^*\right)$. The total loss function is the sum of the loss function for all nodes; namely, we consider the problem

$$(10) \qquad \min_{U \in \mathbb{R}^{n \times r}} \sum_{i \in [n]} f_i\left[(UU^T)_{\mathcal{E}_i}; M_{\mathcal{E}_i}^*\right].$$

Since $\mathcal{E} = \cup_{i \in [n]} \mathcal{E}_i$, the above problem is a special case of the general problem

$$(11) \qquad \min_{U \in \mathbb{R}^{n \times r}} f\left[(UU^T)_{\mathcal{E}}; M_{\mathcal{E}}^*\right],$$

where we define $f\left[(UU^T)_{\mathcal{E}}; M_{\mathcal{E}}^*\right] := \sum_{i \in [n]} f_i\left[(UU^T)_{\mathcal{E}_i}; M_{\mathcal{E}_i}^*\right]$. We name the problem (11) as the *matrix sensing over graphs* (MSoG) problem. The MSoG problem has a number of applications, including the state estimation problem in power systems [39, 24]. We note that in those applications, the loss function $f_i$ is a quadratic function of entries of $M_{\mathcal{E}_i}$ and $M_{\mathcal{E}_i}^*$; see the example in the end of this section and the graph-structured quadratic sensing problem in [24].

To better understand (10) as a special type of MSoG, we consider a toy example where the undirected graph $\mathcal{G}$ has $n = 6$ vertices and is plotted in Figure 1. In this example, the objective function of the MSoG problem is

$$\begin{aligned}
f(M_{\mathcal{E}}; M_{\mathcal{E}}^*) = {} & f_1(M_{11}, M_{12}, M_{13}; M_{11}^*, M_{12}^*, M_{13}^*) + f_2(M_{21}, M_{22}; M_{21}^*, M_{22}^*) \\
& + f_3(M_{31}, M_{34}, M_{35}; M_{31}^*, M_{34}^*, M_{35}^*) + f_4(M_{41}, M_{43}, M_{46}; M_{41}^*, M_{43}^*, M_{46}^*) \\
& + f_5(M_{53}, M_{56}; M_{53}^*, M_{56}^*) + f_6(M_{64}, M_{65}, M_{66}; M_{64}^*, M_{65}^*, M_{66}^*),
\end{aligned}$$

where we recall that we factorize $M$ into $UU^T$ in the Burer-Monteiro factorization approach. Hence, only 13 entries of the matrix $M^*$ with $n^2 = 36$ entries appear in the measurements. For example, the entry $M_{16}^*$ does not appear in any measurements. If we directly observe these 13 entries, we can define the loss function $f_1$ by the $\ell_2$-loss function:

$$f_1(M_{11}, M_{12}, M_{13}; M_{11}^*, M_{12}^*, M_{13}^*) = (M_{11} - M_{11}^*)^2 + (M_{12} - M_{12}^*)^2 + (M_{13} - M_{13}^*)^2.$$

We can define other loss functions $f_2, \ldots, f_6$ in a similar way. Then, the MSoG problem reduces to the matrix completion problem with $\Omega = \mathcal{E}$, namely, the objective function becomes

$$f(M_{\mathcal{E}}; M_{\mathcal{E}}^*) = \|M_{\mathcal{E}} - M_{\mathcal{E}}^*\|_F^2.$$

Therefore, the matrix completion problem (6) is a special case of the MSoG problem (11) and the existing results for the matrix completion problem cannot be directly applied to problem (11). Similarly, if $\Omega$ is a complete graph, the matrix sensing problem (10) becomes a special case of the MSOG problem. Moreover, since some entries of $M^*$ do not appear in any of the measurements in general, it is easy to verify that the RIP condition (5) does not hold for problem (11). In summary, the existing results based on the RIP condition [38] and the incoherence condition [17, 13], as well as the results for other applications of the low-rank matrix optimization problem, cannot be directly applied to the MSoG problem.

Next, we show that the power state estimation problem [39, 24] can be formulated as the MSoG problem (11). This is the most important data-analysis problem for power systems and is solved every 5-15 minutes in practice using heuristic methods with little theoretical guarantees. Several major blackouts were attributed to the failure of solving this problem. An $N$-bus power system can be represented by an undirected graph $\mathcal{G} = (\mathcal{V}, \mathcal{E})$, where $\mathcal{V} = [N]$ and $\mathcal{E} \subset \mathcal{V} \times \mathcal{V}$ are the set of buses and the set of lines, respectively. The state of the power system is characterized by the voltage phasors $\mathbf{v} \in \mathbb{C}^N$, where $\mathbb{C}^N$ is the $N$-dimensional complex space. Our goal is to recover the voltage phasors via some measurements, without loss of generality, we assume that we measure the voltage magnitude at each bus and the active power flows over each line. Suppose that $\mathbf{v} = \mathbf{x} + \mathbf{i}\mathbf{y}$ for $\mathbf{x}, \mathbf{y} \in \mathbb{R}^N$, where $\mathbf{i}$ is the imagnary unit. Define

$$\mathbf{z} := \begin{bmatrix} \mathbf{x} \\ \mathbf{y} \end{bmatrix} \in \mathbb{R}^{2N}.$$

The following two types of measurements are available:

1. For each bus $k \in \mathcal{V}$, the **voltage magnitude** is defined as $|v_k|^2$, where the magnitude for a complex number $v_k = x_k + \mathbf{i}y_k$ is defined as $|v_k|^2 := x_k^2 + y_k^2$. The voltage magnitude can be written in the matrix form as

$$|v_k|^2 = \mathbf{z}^T(\mathbf{e}_k\mathbf{e}_k^T + \mathbf{e}_{k+N}\mathbf{e}_{k+N}^T)\mathbf{z},$$

where $\mathbf{e}_k$ is the $k$-th standard basis of $\mathbb{R}^N$.

2. For each line $(k, \ell) \in \mathcal{E}$, let $Y_{k\ell} \in \mathbb{C}$ be the admittance of the line. The **active power flow** from bus $k$ to bus $\ell$ is $p_{k\ell} = \mathrm{Re}\left(v_k(v_k - v_\ell)^* Y_{k\ell}^*\right)$, where $z^*$ is the conjugate of complex number $z$. We observe the active power flows $p_{k\ell}$ and $p_{\ell k}$ for all lines $(k, \ell) \in \mathcal{E}$. Suppose that $Y_{k\ell} = P_{k\ell} + \mathbf{i}Q_{k\ell}$ for $P_{k\ell}, Q_{k\ell} \in \mathbb{R}$ for all $(k, \ell) \in \mathcal{E}$. Then, the active power flow can be written in the matrix form as

$$p_{k\ell} = \mathbf{z}^T\left[P_{k\ell}(\mathbf{e}_k\mathbf{e}_k^T - \mathbf{e}_k\mathbf{e}_\ell^T + \mathbf{e}_{k+N}\mathbf{e}_{k+N}^T - \mathbf{e}_{N+k}\mathbf{e}_{N+k}^T) + Q_{k\ell}(\mathbf{e}_k\mathbf{e}_{N+\ell}^T - \mathbf{e}_\ell\mathbf{e}_{N+k}^T)\right]\mathbf{z}.$$

Therefore, these observations can be written as the quadratic form $\mathbf{z}^T M_j \mathbf{z}$ for some matrices $M_1, \dots, M_m \in \mathbb{R}^{2N \times 2N}$, where $m = 2N + |\mathcal{E}|$ is the number of observations. For each sensing matrix $M_j$, only the diagonal entries and $(k, \ell), (N+k, \ell), (k, N+\ell), (N+k, N+\ell)$ entries for $(k, \ell) \in \mathcal{E}$ may be nonzero. In addition, we fix $y_1 = z_{N+1} = 0$ since a unitary transformation on $\mathbf{v}$ will not change the loss function. As a result, the problem can be written as a $(2N - 1)$-dimensional rank-1 MSoG problem:

$$\min_{\mathbf{w} \in \mathbb{R}^{2N}} \sum_{j \in [m]} \left\langle M_j, \mathbf{w}\mathbf{w}^T - \mathbf{z}\mathbf{z}^T \right\rangle^2, \quad \text{s.t. } w_{N+1} = 0.$$

## 1.4 Problem formulation and contributions

In this work, we propose the first sufficient condition that guarantees the benign landscape of the MSoG problem. More specifically, we consider the case when the graph $\mathcal{G}$ is a random graph obeying the Erdös–Rényi model. Namely, each pair $(i, j) \in \mathcal{V} \times \mathcal{V}$ belongs to the edge set independently with probability $p$. Under this random graph model and the assumption that the vertex set $\mathcal{V}$ is $[n]$, each entry $M_{ij}^*$ is indirectly observed (i.e., is involved in some measurements) independently with probability $p$. For comparison with the existing results of the matrix completion problem (6), we denote the edge set, which is equivalent to the set of indices of observed entries, as $\Omega \subset [n] \times [n]$. Then, the set $\Omega$ follows the Bernoulli sampling model (Definition 3). Similar to the matrix completion problem, we aim at recovering the ground truth matrix $M^*$ by solving the following problem with an

improved regularizer:

$$(12) \qquad \min_{U \in \mathbb{R}^{n \times r}} \frac{1}{p} f \left[ \left(UU^T\right)_\Omega ; M_\Omega^* \right] + \sum_{i \in [n]} r \left( \|e_i^T U\|_F \right),$$

where we define the regularizer

$$r(x) := \lambda \int_{-1}^{1} \left[ (x + \alpha y - 10\alpha)_+ + 9\alpha \right]^4 (1 - |y|) \ dy, \quad \forall x \in \mathbb{R}.$$

Note that in problem (12), we use a novel incoherence regularizer that is different from those in the prior literature. The regularizer is the convolution between $\left[ (x + \alpha y - 10\alpha)_+ + 9\alpha \right]^4$ and the probability density function $1 - |y|$ on $[-1, 1]$, which is twice continuously differentiable in $x$ (the set of discontinuous points of the derivatives is a zero measure set). Hence, the regularizer $r\left( \|e_i^T U\|_F \right)$ is twice continuously differentiable in $U$. In addition, we can exchange the convolution and the differentiation with respect to $x$. Since the integrand is a quartic polynomial, the objective value and the derivatives of the regularizer can be exactly evaluated by numerical integration schemes with $O(1)$ computations. Let $\mathcal{R} \subset \mathbb{R}^{n \times r}$ be the set of $U$ such that $\|\mathbf{e}_i^T U\|_F = O(\alpha)$ for all $i \in [n]$. The intuition behind the design of the incoherence regularizer is to ensure that every approximate first-order critical point of problem (11) must lie in set $\mathcal{R}$. More specifically, the regularizer ensures that (i) outside $\mathcal{R}$, the gradient of the regularizer is large enough such that any approximate first-order critical point must lie in $\mathcal{R}$, (ii) the regularizer and its Hessian matrix have a sufficiently small contribution to the objective function in $\mathcal{R}$.

Besides the randomness model of $\Omega$, we make the following three assumptions about problem (12).

**Assumption 1.** *The loss function $f$ is twice continuously differentiable.*

**Assumption 2.** *The ground truth matrix $M^*$ is PSD and rank-$r$.*

**Assumption 3.** *The ground truth matrix $M^*$ is a global minimizer of $f[(\cdot)_\Omega; M_\Omega^*]$.*

We note that these assumptions are standard in the low-rank optimization literature. The first assumption is a mild regularity assumption on the loss function and is satisfied by a wide range of loss functions, including the $\ell_2$-loss and the negative log-likelihood function of various probability distributions. The second assumption is based on the prior knowledge about the specific application. The third assumption is necessary for the exact recovery of the ground truth $M^*$.

We give an informal statement of the main result in the following theorem. Here, we consider the $(\delta, \Omega)$-RIP$_{2r,2r}$ condition, which is an extension of the classic $\delta$-RIP$_{2r,2r}$ condition. Intuitively, the constant $\delta \in [0, 1)$ measures the similarity between $f$ and the $\ell_2$-loss function. The rigorous definition is provided in Definition 4.

**Theorem 1** (Informal). *Suppose that the loss function $f$ satisfies the $(1/16, \Omega)$-RIP$_{2r,2r}$ condition with a high probability over $\Omega$ and the sampling rate $p$ satisfies*

$$np \ge \Theta[\mu^2 r^3 (\kappa^*)^2 \log n],$$

*where $\kappa^*$ is the condition number of $M^*$. Then, with a suitable choice of the parameters $\alpha$ and $\lambda$, problem (12) satisfies the strict saddle property with high probability. Furthermore, there exist algorithms that can find a solution $U_0$ such that $\|U_0 U_0^T - M^*\|_F \le \epsilon$ in polynomial time with the same probability.*

The above theorem provides the first theoretical result for the MSoG problem. Basically, Theorem 1 says that a combination of the incoherence condition and the $\Omega$-RIP condition is sufficient to guarantee that problem (12) can be solved in polynomial time with high probability. In addition, the lower bound on the sampling rate is better than the bound in [17] (i.e., the bound in (9)). This is a result of our improved regularizer. Finally, the upper bound on the $\Omega$-RIP constant is an absolute constant (namely, $1/16$).

**Remark 1.** *We note that since the result of Theorem 1 only holds with high probability, it is not necessary for the $\Omega$-RIP condition to hold for all subsets $\Omega$. Instead, we only require the $\Omega$-RIP condition to hold for a set of $\Omega$ that correspond to a high probability over the distribution of $\Omega$. In the special case when $\Omega$ is a deterministic subset that satisfies similar "benign" properties as in the random graph case, we only need the $\Omega$-RIP condition for this deterministic subset $\Omega$.*

The next informal theorem shows that our upper bound is optimal up to a constant.

**Theorem 2** (Informal)**.** *There exists a loss function $f$ such that: (i) $f$ satisfies the $(1/2, \Omega)$-RIP$_{2r,2r}$ condition for all $\Omega \subset [n] \times [n]$, (ii) for all $p \in [0,1]$, problem* (12) *has a spurious second-order critical point[2] with probability at least $(3 - \sqrt{5})/2 \approx 0.38$.*

Intuitively, Theorem 2 provides a negative result saying that the tightest upper bound on the RIP constant cannot be better than $1/2$ if the problem (12) has a benign landscape with high probability. This is also the first negative result on the problem (12).

## 1.5 Notation

For every natural number $n$, we denote $[n] := \{1, \ldots, n\}$. The operator 2-norm and the Frobenius norm of a matrix $M$ are denoted as $\|M\|_2$ and $\|M\|_F$, respectively. The trace of matrix $M$ is denoted as $\mathrm{tr}(M)$. The inner product between two matrices is defined as $\langle M, N \rangle := \mathrm{tr}(M^T N)$. For any matrix $M \in \mathbb{R}^{n \times n}$, we denote its singular values by $\sigma_1(M) \geq \cdots \geq \sigma_k(M)$. Let $\sigma_i^*$ be the singular value $\sigma_i(M^*)$. The condition number of $M^*$ is $\kappa^* = \sigma_1^*/\sigma_r^*$. The minimum eigenvalue of matrix $M$ is denoted as $\lambda_{min}(M)$. For any two matrices $A, B \in \mathbb{R}^{n \times m}$, we use $A \otimes B$ to denote the fourth-order tensor whose $(i, j, k, \ell)$ element is $A_{i,j} B_{k,\ell}$. The identity tensor is denoted as $\mathcal{I}$. The notation $M \succeq 0$ means that the matrix $M$ is PSD. The sub-matrix $R_{i:j;k:\ell}$ consists of the $i$-th to the $j$-th rows and the $k$-th to the $\ell$-th columns of matrix $R$. The action of the Hessian $\nabla^2 f(M)$ on any two matrices $K$ and $L$ is given by $[\nabla^2 f(M)](K, L) := \sum_{i,j,k,\ell} [\nabla^2 f(M)]_{i,j,k,\ell} K_{ij} L_{k,\ell}$. The notation $f = O(g)$ means that there exists an absolute constant $C > 0$ such that $f \leq C \cdot g$. The notation $f = \Theta(g)$ means that there exist absolute constants $C_1, C_2, > 0$ such that $C_1 \cdot g \leq f \leq C_2 \cdot g$.

## 2 $\Omega$-RIP Condition

Since the objective function $f(M_\Omega; M_\Omega^*)$ does not satisfy the classic RIP condition unless all elements of $M^*$ are observed (i.e., when $\Omega = [n] \times [n]$), it is necessary to consider a generalization of the RIP condition to the partial observation case.

**Definition 4** ($\Omega$-RIP Condition)**.** *Given a subset $\Omega \subset [n] \times [n]$ and natural numbers $r, s$, the function $f(\cdot; M^*)$ is said to satisfy the $\Omega$-**Restricted Isometry Property** ($\Omega$-RIP) of rank $(2r, 2s)$ for a constant $\delta \in [0, 1)$, denoted as $(\delta, \Omega)$-RIP$_{2r,2s}$, if*

$$(13) \qquad (1 - \delta)\|K_\Omega\|_F^2 \leq \left[\nabla^2 f[M_\Omega; (M^*)_\Omega]\right](K_\Omega, K_\Omega) \leq (1 + \delta)\|K_\Omega\|_F^2$$

*holds for all matrices $M, K \in \mathbb{R}^{n \times n}$ such that $\mathrm{rank}(M) \leq 2r, \mathrm{rank}(K) \leq 2s$.*

Note that the matrix completion problem satisfies the $(0, \Omega)$-RIP$_{2r,2s}$ condition. In the following, we demonstrate another example for which the $\Omega$-RIP condition holds with a non-zero constant $\delta$.

**Example 1** (Linear matrix sensing over graphs)**.** *In the linear matrix sensing over graphs problem, the loss function is defined as*

$$f(M_\Omega; M_\Omega^*) := \|\mathcal{A}(M_\Omega) - \mathcal{A}(M_\Omega^*)\|_F^2,$$

*where the linear operator $\mathcal{A} : \mathbb{R}^{n \times n} \mapsto \mathbb{R}^m$ (defined in* (3)*) is generated by Gaussian measurements and $m$ is the number of measurements. For all subsets $\Omega \subset [n] \times [n]$, a similar proof to that of Theorem 2.3 of [9] implies that the $(\delta, \Omega)$-RIP$_{2r,2r}$ condition holds with high probability when $m \geq cnr/\delta^2$ for some constant $c > 0$. The intuition behind the proof is that the constructed $\epsilon$-net for the linear matrix sensing problem is also an $\epsilon$-net for the linear MSoG problem since $\|M_\Omega - M_\Omega'\|_F \leq \|M - M'\|_F$ for all $M, M' \in \mathbb{R}^{n \times n}$.*

**Remark 2.** *In the above example and other examples in practice, the $\Omega$-RIP condition holds with high probability for a fixed subset $\Omega$. Therefore, we can focus on the event when the $\Omega$-RIP condition holds since otherwise the results will only differ by a sufficiently small probability.*

In Section 4 and Appendix C, we numerically show that the optimization complexity of problem (11) is closely related to the $\Omega$-RIP constant $\delta$ and the sampling rate $p$.

---

[2]A point $U_0$ is called a spurious second-order critical point of problem (12) if $U_0$ satisfies the first-order and the second-order necessary optimality conditions but $U_0 U_0^T \neq M^*$.

# 3 Theoretical Results

In this section, we provide strong theoretical results on the MSoG problem (12). We first develop a sufficient condition on the benign landscape of problem (12) and then study the tightness of our developed condition.

## 3.1 Global landscape: Strict saddle property

First, we develop conditions under which problem (12) does not have any spurious second-order critical points and therefore saddle-escaping methods (e.g., [22, 23]) can find an approximate global minimum in polynomial time. To guarantee the global convergence, the strict saddle property is commonly considered in the literature:

**Definition 5** (Strict Saddle Property [33]). *Consider an optimization problem $\min_{x \in \mathcal{X} \subset \mathbb{R}^d} F(x)$ and let $\mathcal{X}^*$ denote the set of its global minima. We say that the problem satisfies the $(\theta, \beta, \gamma)$-**strict saddle property** for $\theta, \beta, \gamma > 0$ if at least one of the following conditions is satisfied for every $x \in \mathcal{X}$:*

$$\text{dist}(x, \mathcal{X}^*) \leq \theta; \quad \|\nabla F(x)\|_F \geq \beta; \quad \lambda_{min}[\nabla^2 F(x)] \leq -\gamma,$$

*where $\text{dist}(x, \mathcal{X}^*) := \inf_{x^* \in \mathcal{X}^*} \|x - x^*\|_F$ is the distance between $x$ and $\mathcal{X}^*$.*

For the low-rank optimization problem, the distance in the factorization space is equivalent to the distance in the matrix space in the sense that there exist constants $c_1(\mathcal{X}^*), c_2(\mathcal{X}^*) > 0$ such that

$$c_1(\mathcal{X}^*) \cdot \|U - U^*\|_F \leq \|UU^T - U^*(U^*)^T\|_F \leq c_2(\mathcal{X}^*) \cdot \|U - U^*\|_F$$

holds for all $U \in \mathcal{X}$ as long as $\|U - U^*\|_F$ is small and $\mathcal{X}^*$ is bounded [34]. Denote the objective function of problem (12) as $\ell(U)$. As an example of saddle-avoiding methods, the accelerated perturbed gradient descent algorithm [23] can find a point $U_0$ such that

$$\|\nabla \ell(U_0)\|_F = O(\epsilon), \quad \lambda_{min}[\nabla^2 \ell(U_0)] = -O(\sqrt{\epsilon})$$

in $O(\epsilon^{-1.75})$ iterations with high probability. If we choose $\epsilon > 0$ to be small enough, the strict saddle property ensures that the point $U_0$ satisfies $\|U_0 U_0^T - M^*\|_F = O(\theta)$. We note that the Lipschitz continuity of the Hessian of $\ell$ can be guaranteed by the regularity assumption 1 and the boundedness of trajectories of the algorithm, which can be proved in a similar way as Theorem 8 in [22]. In summary, if the strict saddle property holds, we can apply saddle-avoiding methods to achieve the polynomial-time global convergence for problem (12).

Now, we prove that the problem (12) satisfies the strict saddle property with high probability under the $\Omega$-RIP$_{2r,2r}$ condition and the incoherence condition. Compared with Theorem 1, we assume, for the simplicity of the statement of the theorem, that the $\Omega$-RIP condition holds for all subset $\Omega$ since otherwise, the result will only differ by a small probability.

**Theorem 3.** *Suppose that the loss function $f$ satisfies the $(\delta, \Omega)$-RIP$_{2r,2r}$ condition for all $\Omega \subset [n] \times [n]$ and*

$$\alpha^2 = \Theta\left(\frac{\mu r \sigma_1^*}{n}\right), \quad \lambda = \Theta\left[\frac{(\sqrt{\mu} + \sqrt{n}\delta)n}{\sqrt{\mu}r}\right], \quad np \geq \Theta[\mu^2 r^3 (\kappa^*)^2 \log n], \quad \delta < \frac{1}{16}.$$

*Then, there exists a small constant $\epsilon > 0$ such that with probability at least $1 - 1/\text{poly}(n)$, the MSoG problem (12) satisfies the $(\theta, \beta, \gamma)$-strict saddle property with*

$$\theta = \Theta(\epsilon/\sigma_r^*), \quad \beta = \epsilon, \quad \gamma = \Theta(\epsilon^2/\sigma_r^*).$$

In Theorem 3, we provide the first theoretical results on the MSoG problem (12). Basically, the theorem shows that if the $\Omega$-RIP$_{2r,2r}$ constant is smaller than an absolute constant, then the same sampling rate as for the matrix completion problem is sufficient to guarantee the benign landscape of problem (12). Therefore, the result is a generalization of the results in [17], which considered the case when the $\Omega$-RIP constant $\delta$ is zero and the sampling rate $p$ satisfies a stricter condition (9). On the other hand, if the sampling rate $p$ is equal to 1, Theorem 3 guarantees that the strict saddle property holds when the (regular) RIP condition holds with $\delta < 1/16$. Compared with the state-of-the-art results in [38, 5], the upper bound on $\delta$ in our work may not be optimal but is only worse by an absolute constant. We leave the improvement of the upper bound as future work.

## 3.2 Tightness of the $\Omega$-RIP condition

To study the tightness of our $\Omega$-RIP condition for problem (12), we construct an instance of problem (12) that satisfies the $(1/2, \Omega)$-RIP condition but has spurious second-order critical points. Note that the existence of spurious second-order critical points negates the strict saddle property. The counterexample is based on a similar idea as those in [38]. More specifically, we assume that $n \geq 2r$ and consider the tensor:

$$
\begin{aligned}
\mathcal{H} := \sum_{i \in [r]} \Big\{ &- \frac{1}{2} \left[ (e_{2i-1}e_{2i-1}^T) \otimes (e_{2i-1}e_{2i-1}^T) + (e_{2i}e_{2i}^T) \otimes (e_{2i}e_{2i}^T) \right] \\
&+ \frac{1}{2} \left[ (e_{2i-1}e_{2i-1}^T) \otimes (e_{2i}e_{2i}^T) + (e_{2i}e_{2i}^T) \otimes (e_{2i-1}e_{2i-1}^T) \right] \\
&- \frac{1}{4} \left[ (e_{2i-1}e_{2i}^T) \otimes (e_{2i-1}e_{2i}^T) + (e_{2i}e_{2i-1}^T) \otimes (e_{2i}e_{2i-1}^T) \right] \\
&+ \frac{1}{4} \left[ (e_{2i-1}e_{2i}^T) \otimes (e_{2i}e_{2i-1}^T) + (e_{2i}e_{2i-1}^T) \otimes (e_{2i-1}e_{2i}^T) \right] \Big\},
\end{aligned}
$$

where $e_i \in \mathbb{R}^n$ is the $i$-th standard basis of $\mathbb{R}^n$. The rank-$r$ ground truth matrix $M^*$ is constructed as

$$
U^* := [e_1 \quad e_3 \quad \cdots \quad e_{2r-1}], \quad M^* := U^*(U^*)^T = \sum_{i \in [r]} e_{2i-1}e_{2i-1}^T.
$$

Then, the loss function is given by

$$
f_{3/2}(M_\Omega; M_\Omega^*) := \frac{1}{2}(M_\Omega - M_\Omega^*) : \left( \frac{3}{2} \cdot \mathcal{I} + \mathcal{H} \right) : (M_\Omega - M_\Omega^*), \quad \forall M \in \mathbb{R}^{n \times n}.
$$

It is proved in [38] that if $\Omega = [n] \times [n]$, the function $f_{3/2}$ satisfies the $1/2$-RIP$_{2r,2r}$ condition and has a spurious second-order critical point. In this work, we generalize the results to the case when the set $\Omega$ is random and the objective function contains a regularizer.

**Theorem 4.** *Suppose that the loss function in problem* (12) *is chosen to be* $f_{3/2}$. *Then, it holds that:*

1. *The loss function* $f_{3/2}$ *satisfies the* $(1/2, \Omega)$-*RIP*$_{2r,2r}$ *condition for all* $\Omega \subset [n] \times [n]$;

2. *For all* $p \in [0, 1]$, *problem* (12) *has a spurious second-order critical point with probability at least* $\max\{p^{r(r+1)}, 1 - p^{r(r+1)/2}\} \geq (3 - \sqrt{5})/2$.

We note that the results in Theorem 4 holds for all $p \in [0, 1]$. From Theorem 4, we cannot improve the upper bound of $\delta$ in Theorem 3 to be better than $1/2$. In addition, this result shows that our bound in Theorem 3 is optimal up to a constant. This tightness result is consistent with that of the matrix sensing problem in [40, 38].

# 4 Numerical Illustrations

In this section, we show how the optimization complexity is related to the $\Omega$-RIP constant $\delta$ and the sampling rate $p$ via a numerical example. Here, the optimization complexity refers to the probability that the randomly initialized gradient descent algorithm can find the ground truth matrix $M^*$. In this example, we choose a random orthogonal matrix $V \in \mathbb{R}^{n \times n}$ and define the loss function to be

$$
f_c[M_\Omega; (VM^*V^T)_\Omega] := \frac{1}{2}[M - (VM^*V^T)]_\Omega : (c \cdot \mathcal{I} + \mathcal{H}) : [M - (VM^*V^T)]_\Omega, \forall M \in \mathbb{R}^{n \times n},
$$

where $c \in \mathbb{R}$ is a hyper-parameter and the tensor $\mathcal{H}$ and the ground truth $M^*$ are defined in Section 3.2. In addition, by a similar analysis as in Section 3.2, we can prove that the function $f_c$ satisfies the $(1/2, \Omega)$-RIP$_{2r,2r}$ condition if we choose $c = 3/2$. We also numerically verify this conclusion by checking the curvature $[\nabla^2 f_{3/2}(M_\Omega; M_\Omega^*)](K, K)$ along $10^4$ random directions $K \in \mathbb{R}^{n \times n}$. Since $f_{3/2}$ is a quadratic function, the $(1/2, \Omega)$-RIP$_{2r,2r}$ condition is given by

$$
\frac{1}{2}\|K_\Omega\|_F^2 \leq K_\Omega : \left( \frac{3}{2} \cdot \mathcal{I} + \mathcal{H} \right) : K_\Omega \leq \frac{3}{2}\|K_\Omega\|_F^2, \quad \forall K \in \mathbb{R}^{n \times n},
$$

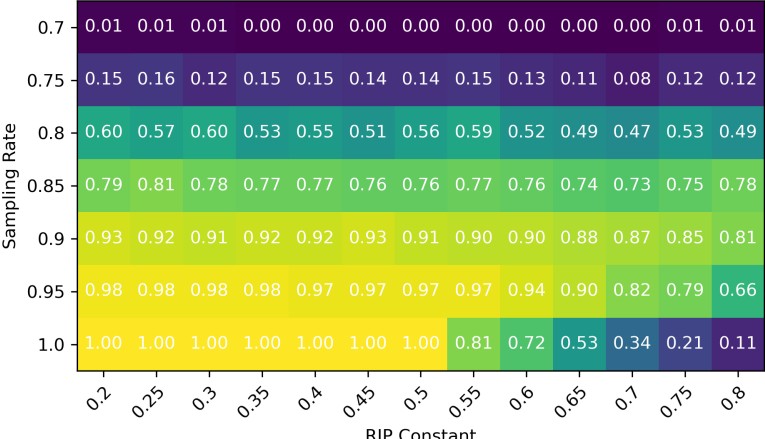

Figure 2: Success rate of the algorithm with different sampling rate $p$ and RIP constant $\delta$.

where we define the tensor multiplication $K : \mathcal{H}' : K = \sum_{i,j,k,\ell \in [n]} \mathcal{H}'_{ijk\ell} K_{ij} K_{k\ell}$ for all $K \in \mathbb{R}^{n \times n}$ and fourth-order tensor $\mathcal{H}' \in \mathbb{R}^{n \times n \times n \times n}$. In addition, since the identity tensor $\mathcal{I}$ satisfies the $(0, \Omega)$-RIP$_{2r,2r}$ condition, it is straightforward to prove that the function $f_c$ satisfies the $(\delta, \Omega)$-RIP$_{2r,2r}$ condition with $\delta = 1/(2c-1)$ for all $c \geq 1$.

We choose the problem size to be $n = 10$ and $r = 5$. The regularization parameters are $\alpha = 10$ and $\lambda = 100$. The set of sampling rates and $\Omega$-RIP$_{2r,2r}$ constants are

$$p \in \{0.7, 0.75, \ldots, 0.95, 1.0\}, \quad \delta \in \{0.2, 0.25, \ldots, 0.75, 0.8\}.$$

We solve each problem instance by the Burer-Monterio factorization and the perturbed accelerated gradient descent algorithm [23], where the constant step size is $0.007/c$. We generate 100 independent problem instances and compute the success rate of the gradient descent algorithm with random initialization. We say that the algorithm successfully solves the instance if the generalization error $\|UU^T - M^*\|_F$ is less than $10^{-3}$. If this condition fails, it means that the algorithm is stuck at a spurious local solution.

The results are plotted in Figure 2. We can see from the figure that the optimization complexity grows when $\delta$ becomes smaller and when $p$ becomes larger. This result shows that the $\Omega$-RIP condition plays an important role in characterizing the optimization complexity of problem (12). To be more concrete, we expect that an upper bound on the $\Omega$-RIP constant will be able to guarantee the benign optimization landscape of problem (12). Moreover, the result is consistent with the results for the matrix sensing problem and the matrix completion problem. Furthermore, we can see that when the $\Omega$-RIP constant is smaller than $1/2$, the success rate has a stronger correlation with $p$ than $\delta$. This observation is also reflected in Theorem 1, where the lower bound on the sampling rate is on the same order as those in the existing works [18, 17] if $\delta$ is upper bounded. For the cases when $p$ is close to 1 and $\delta$ is larger than 0.5, the success rate is better than the case when $p = 1$. One possible explanation for this phenomenon is that the loss function $f_c$ has multiple global minima when the sampling rate is smaller than 1. Instead of converging to a spurious local solution, the implicit regularization [26] makes the perturbed gradient descent algorithm more likely to converge to the global solution with the "minimal complexity", which is likely the ground truth $VM^*V^T$. If the sampling rate is much smaller than 1, the number of spurious local minima will increase and the perturbed gradient descent algorithm may stuck at spurious local minima and fail to converge to the ground truth.

## 5 Conclusion

In this work, we provide the first theoretical analysis of the MSoG problem. A new notion, dubbed as the $\Omega$-RIP condition, is proposed and shown to be useful in characterizing the optimization complexity of the MSoG problem. Using an improved incoherence regularizer, we proved the polynomial-time global convergence of saddle-avoiding methods under the $\Omega$-RIP condition and the incoherence condition. The bounds on the sampling rate and the $\Omega$-RIP constant are state-of-the-art up to a constant. Moreover, we showed that our bound on the $\Omega$-RIP condition is tight (up to a constant). Future works include improving the upper bound on the $\Omega$-RIP constant and the lower bound on the sampling rate.

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

## Acknowledgements

This work was supported by grants from AFOSR, ARO, ONR, and NSF. Haixiang Zhang was supported by the Two Sigma Ph.D. Fellowship.

