# Appendix

Without loss of generality, we assume that the loss function is symmetric in $M$, namely,

$$f(M; M^*) = f(M^T; M^*), \quad \forall M \in \mathbb{R}^{n \times n}.$$

Otherwise, we can replace the loss function by $\bar{f}(M; M^*) := [f(M; M^*) + f(M^T; M^*)]/2$ and this will not change the values of the objective function in problem (1) in the feasible set.

Denote the objective function of problem (12) as $\frac{1}{p}h(U) + g(U)$, where

$$h(U) := f[(UU^T)_\Omega; M_\Omega^*], \quad g(U) := \sum_{i \in [n]} r(\|e_i^T U\|_F).$$

For every $\Delta \in \mathbb{R}^{n \times r}$, the gradient and the Hessian matrix of $h(U)$ satisfy

$$\langle \nabla h(U), \Delta \rangle = 2 \left\langle \nabla f[(UU^T)_\Omega; M_\Omega^*], \left(U\Delta^T\right)_\Omega \right\rangle,$$

$$\left[\nabla^2 h(U)\right](\Delta, \Delta) = 2 \left\langle \nabla f[(UU^T)_\Omega; M_\Omega^*], \left(\Delta\Delta^T\right)_\Omega \right\rangle$$
$$+ \left[\nabla^2 f[(UU^T)_\Omega; M_\Omega^*]\right] \left[\left(\Delta U^T + U\Delta^T\right)_\Omega, \left(\Delta U^T + U\Delta^T\right)_\Omega\right].$$

Since we can exchange the convolution and the derivatives, the gradient and the Hessian matrix of $g(U)$ satisfy

$$\langle \nabla g(U), \Delta \rangle = 4\lambda \int_{-1}^1 \sum_{i \in [n]} \left[\left(\|e_i^T U\|_F + \alpha y - 10\alpha\right)_+ + 8\alpha\right]^3 \frac{e_i^T U \Delta^T e_i}{\|e_i^T U\|_F}(1 - |y|) \, dy,$$

$$\left[\nabla^2 g(U)\right](\Delta, \Delta)$$
$$= 4\lambda \int_{-1}^1 \sum_{i \in [n]} \left[\left(\|e_i^T U\|_F + \alpha y - 10\alpha\right)_+ + 8\alpha\right]^3 \frac{\|e_i^T U\|_F \|e_i^T \Delta\|_F - e_i^T U \Delta^T e_i}{\|e_i^T U\|_F^3}(1 - |y|) \, dy$$
$$+ 12\lambda \int_{-1}^1 \sum_{i \in [n]} \left[\left(\|e_i^T U\|_F + \alpha y - 10\alpha\right)_+ + 8\alpha\right]^2 \left(\frac{e_i^T U \Delta^T e_i}{\|e_i^T U\|_F}\right)^2 (1 - |y|) \, dy.$$

For all $\epsilon > 0$, we say that a point $U \in \mathbb{R}^{n \times r}$ is an $\epsilon$-*approximate first-order critical point* of problem (12) if

$$\left\|\frac{1}{p}\nabla h(U) + \nabla g(U)\right\|_F \leq \epsilon.$$

## A  Proof of Theorem 3

We first bound the norm of an approximate first-order critical point.

**Lemma 1.** *Suppose that $U$ is an $\epsilon$-approximate first-order critical point of problem* (12) *for a sufficiently small $\epsilon > 0$ and*

$$\delta < 1, \quad \max_{i \in [n]} \|e_i^T U\|_F \geq 11\alpha.$$

*Then, it holds that*

$$\|(UU^T)_\Omega\|_F \leq \frac{1 + \delta}{1 - \delta}\|(M^*)_\Omega\|_F.$$

*Proof.* Assume that $U$ is an $\epsilon$-approximate first-order critical point such that

(14) $$\|(UU^T)_\Omega\|_F > \frac{1 + \delta}{1 - \delta}\|(M^*)_\Omega\|_F.$$

Using the approximate first-order stationarity, we have

$$\epsilon\|U\|_F \geq \left\langle \left[\frac{1}{p}\nabla h(U) + \nabla r(U)\right], U \right\rangle = \left\langle \frac{1}{p}\nabla h(U), U \right\rangle + \langle \nabla g(U), U \rangle.$$

For the first term, we can calculate that

$$
\left\langle \frac{1}{p}\nabla h(U), U \right\rangle = \frac{2}{p}\left\langle \nabla f[(UU^T)_\Omega], (UU^T)_\Omega \right\rangle
$$

$$
= \frac{2}{p}\int_0^1 \left[\nabla^2 f[(M^*)_\Omega + t(UU^T - M^*)_\Omega; (M^*)_\Omega]\right][(UU^T)_\Omega, (UU^T - M^*)_\Omega]\, dt
$$

$$
\geq \frac{2(1-\delta)}{p}\left\|(UU^T)_\Omega\right\|_F^2 - \frac{2(1+\delta)}{p}\|(UU^T)_\Omega\|_F\|(M^*)_\Omega\|_F \geq 0,
$$

where the first inequality is from the $(\delta, \Omega)$-RIP$_{2r,2r}$ condition and Lemma 11 of [4], and the last inequality is from the assumption (14). Define the set

$$
\mathcal{I} := \{i \in [n] \mid \|e_i^T U\|_F \geq 11\alpha\}.
$$

Then, for the second term, we have

$$
\langle \nabla g(U), U \rangle = 4\lambda \int_{-1}^1 \sum_{i\in[n]} \left[\left(\|e_i^T U\|_F + \alpha y - 10\alpha\right)_+ + 8\alpha\right]^3 \frac{e_i^T UU^T e_i}{\|e_i^T U\|_F}(1 - |y|)\, dy
$$

$$
= 4\lambda \int_{-1}^1 \sum_{i\in[n]} \left[\left(\|e_i^T U\|_F + \alpha y - 10\alpha\right)_+ + 8\alpha\right]^3 \|e_i^T U\|_F(1 - |y|)\, dy
$$

$$
\geq 4\lambda \int_{-1}^1 \sum_{i\in[n]} \left[\left(\|e_i^T U\|_F - 11\alpha\right)_+ + 8\alpha\right]^3 \|e_i^T U\|_F(1 - |y|)\, dy
$$

$$
\geq 4\lambda \int_{-1}^1 \sum_{i\in\mathcal{I}} \left(\|e_i^T U\|_F - 3\alpha\right)^3 \|e_i^T U\|_F(1 - |y|)\, dy
$$

$$
= 4\lambda \sum_{i\in\mathcal{I}} \left(\|e_i^T U\|_F - 3\alpha\right)^3 \|e_i^T U\|_F.
$$

Combining the last two inequalities, we obtain

$$
\epsilon\|U\|_F \geq 4\lambda \sum_{i\in\mathcal{I}} \left(\|e_i^T U\|_F - 3\alpha\right)^3 \|e_i^T U\|_F.
$$

Define

$$
i^* := \arg\max_{i\in[n]} \|e_i^T U\|.
$$

Then, we obtain

$$
\epsilon\sqrt{n}\|e_{i^*}^T U\|_F \geq \epsilon\|U\|_F \geq 4\lambda \sum_{i\in\mathcal{I}} \left(\|e_i^T U\|_F - 3\alpha\right)^3 \|e_i^T U\|_F \geq 4\lambda \left(\|e_{i^*}^T U\|_F - 3\alpha\right)^3 \|e_{i^*}^T U\|_F,
$$

which further leads to

$$
\left(\|e_{i^*}^T U\|_F - 3\alpha\right)^3 \leq \frac{\epsilon\sqrt{n}}{4\lambda}.
$$

By choosing $\epsilon$ to be sufficiently small, the above inequality implies that

$$
\|e_{i^*}^T U\|_F < 11\alpha.
$$

This is a contradiction to the condition in the lemma. $\qquad\square$

Next, we provide a generalization to Lemma 9 of [17].

**Lemma 2.** *Suppose that $U$ is an $\epsilon$-approximate first-order critical point of problem (12) for a sufficiently small $\epsilon > 0$ and*

$$
\alpha^2 = \Theta\left(\frac{\mu r \sigma_1^*}{n}\right), \quad \lambda = \Theta\left[\frac{(1-\delta)(\sqrt{\mu} + \sqrt{n}\delta)n}{\sqrt{\mu}r}\right], \quad np \geq \Theta\left(\mu r \log n\right), \quad \delta < 1.
$$

*Then, there exists a constant $c > 0$ such that it holds with probability at least $1 - 1/\mathrm{poly}(n)$ that*

$$
\max_{i\in[n]} \|e_i^T U\|_F^2 = O\left(\frac{\mu r \sigma_1^*}{n}\right).
$$

*Proof.* Let
$$i^* := \arg\max_{i \in [n]} \|e_i^T U\|.$$

We only need to consider the case when $\|e_{i^*}^T U\|_F \geq 11\alpha$. Since $U$ is an $\epsilon$-approximate first-order critical point, it holds that

$$(15) \qquad \epsilon \|e_{i^*}^T U\|_F \geq \left\langle e_{i^*}^T \left[ \frac{1}{p} \nabla h(U) + \nabla g(U) \right], e_{i^*}^T U \right\rangle.$$

Using Taylor's expansion, it holds that

$$\left\langle e_{i^*}^T \cdot \frac{1}{p} \nabla h(U), e_{i^*}^T U \right\rangle$$
$$= \frac{2}{p} \int_0^1 \left[ \nabla^2 f[(M^*)_\Omega + t(UU^T - M^*)_\Omega; (M^*)_\Omega] \right] [(UU^T e_{i^*} e_{i^*}^T)_\Omega, (UU^T - M^*)_\Omega] \, dt.$$

By the $(\delta, \Omega)$-RIP$_{2r,2r}$ condition and Lemma 11 of [4], one can write

$$(16) \quad \left\langle e_{i^*}^T \cdot \frac{1}{p} \nabla h(U), e_{i^*}^T U \right\rangle$$
$$\geq \frac{2}{p} \left\langle (UU^T e_{i^*} e_{i^*}^T)_\Omega, (UU^T - M^*)_\Omega \right\rangle - \frac{2\delta}{p} \|(UU^T e_{i^*} e_{i^*}^T)_\Omega\|_F \left( \|(UU^T)_\Omega\|_F + \|(M^*)_\Omega\|_F \right)$$
$$= \frac{2}{p} \left\langle e_{i^*}^T (UU^T)_\Omega, e_{i^*}^T (UU^T - M^*)_\Omega \right\rangle - \frac{2\delta}{p} \|e_{i^*}^T (UU^T)_\Omega\|_F \left( \|(UU^T)_\Omega\|_F + \|(M^*)_\Omega\|_F \right)$$
$$\geq -\frac{2}{p} \|e_{i^*}^T (UU^T)_\Omega\|_F \|e_{i^*}^T (M^*)_\Omega\|_F - \frac{2\delta}{p} \|e_{i^*}^T (UU^T)_\Omega\|_F \left( \|(UU^T)_\Omega\|_F + \|(M^*)_\Omega\|_F \right).$$

By Lemmas 35 and 39 of [17] and Lemma 1, it holds with probability at least $1 - 1/\text{poly}(n)$ that

$$\frac{1}{\sqrt{p}} \|e_{i^*}^T (M^*)_\Omega\|_F \leq \sqrt{1+\nu} \|e_{i^*}^T M^*\|_F \leq \sqrt{\frac{(1+\nu)\mu r}{n}} \sigma_1^*,$$
$$\frac{1}{\sqrt{p}} \|e_{i^*}^T (UU^T)_\Omega\|_F \leq O(\sqrt{n}) \|UU^T\|_\infty,$$
$$\frac{1}{\sqrt{p}} \left( \|(UU^T)_\Omega\|_F + \|(M^*)_\Omega\|_F \right) \leq \frac{2}{(1-\delta)\sqrt{p}} \|(M^*)_\Omega\|_F \leq \frac{2\sqrt{(1+\nu)r}\sigma_1^*}{1-\delta},$$

where the constant $\nu > 0$ can be made sufficiently small by choosing a large constant in the condition $np \geq \Theta(\mu r \log n)$. Substituting into the inequality (16), it follows that

$$\left\langle e_{i^*}^T \cdot \frac{1}{p} \nabla h(U), e_{i^*}^T U \right\rangle \geq -O(\sqrt{\mu r}\sigma_1^*)\|UU^T\|_\infty - O[\sqrt{nr}\sigma_1^*\delta/(1-\delta)] \cdot \|UU^T\|_\infty$$
$$= -O[(\sqrt{\mu} + \sqrt{n}\delta)\sqrt{r}\sigma_1^*/(1-\delta)] \cdot \|e_{i^*}^T U\|_F^2,$$

where the last equality is from $\|UU^T\|_\infty = \|e_{i^*}^T U\|_F^2$. Additionally, since $\|e_{i^*}^T U\|_F \geq 9\alpha$, we have

$$\left\langle e_{i^*}^T \nabla h(U), e_{i^*}^T U \right\rangle$$
$$= 4\lambda \int_{-1}^1 \left\langle e_{i^*}^T \cdot \sum_{i \in [n]} \left[ \left( \|e_i^T U\|_F + \alpha y - 10\alpha \right)_+ + 8\alpha \right]^3 \frac{e_i e_i^T U}{\|e_i^T U\|_F}, e_{i^*}^T U \right\rangle (1 - |y|) \, dy$$
$$= 4\lambda \int_{-1}^1 \left( \|e_{i^*}^T U\|_F + \alpha y - 2\alpha \right)^3 \|e_{i^*}^T U\|_F (1 - |y|) \, dy$$
$$\geq 4\lambda \left( \|e_{i^*}^T U\|_F - 3\alpha \right)^3 \|e_{i^*}^T U\|_F \geq 4\lambda \cdot \left( \frac{8}{11} \right)^3 \|e_{i^*}^T U\|_F^4 > \lambda \|e_{i^*}^T U\|_F^4.$$

Substituting the last two bounds into inequality (15), we know that

$$\epsilon \|e_{i^*}^T U\|_F \geq \lambda \|e_{i^*}^T U\|_F^4 - O[(\sqrt{\mu} + \sqrt{n}\delta)\sqrt{r}\sigma_1^*/(1-\delta)] \cdot \|e_{i^*}^T U\|_F^2$$

holds with high probability. We can rewrite the above bound as

$$\lambda \|e_{i^*}^T U\|_F^3 \leq \epsilon + \frac{c(\sqrt{\mu} + \sqrt{n}\delta)\sqrt{r}\sigma_1^*}{1 - \delta} \cdot \|e_{i^*}^T U\|_F,$$

where $c > 0$ is a constant. If we choose $\epsilon$ sufficiently small such that

$$\epsilon^{2/3} \leq \frac{c(\sqrt{\mu} + \sqrt{n}\delta)\sqrt{r}\sigma_1^*}{\lambda^{1/3}(1 - \delta)},$$

the inequality leads to

$$\|e_{i^*}^T U\|_F^2 \leq \frac{c(\sqrt{\mu} + \sqrt{n}\delta)\sqrt{r}\sigma_1^*}{\lambda(1 - \delta)}.$$

By our choice of $\lambda$, we conclude that

$$\|e_{i^*}^T U\|_F^2 \leq O\left(\frac{\mu r \sigma_1^*}{n}\right).$$

This concludes our proof. $\qquad\qquad\square$

Now, we consider the second-order necessary optimality condition for an approximate first-order critical point $U$. The next lemma bounds the curvature along the direction $\Delta$.

**Lemma 3.** *For all $U \in \mathbb{R}^{n \times r}$, we define the direction $\Delta := U - U^* R$, where $U^* \in \mathbb{R}^{n \times r}$ satisfies $U^*(U^*)^T = M^*$ and*

$$R \in \underset{S \in \mathbb{R}^{r \times r}, SS^T = I_r}{\arg\min} \|U - U^* S\|_F.$$

*Then, it holds that*

$$\begin{aligned}
\left[\nabla^2 h(U)\right](\Delta, \Delta) \leq{}& 4\langle \nabla h(U), \Delta \rangle - [3 - (5 + 3t)\delta] \cdot \|(M - M^*)_\Omega\|_F^2 \\
&+ [1 + (1 + 3t^{-1})\delta] \cdot \|(\Delta\Delta^T)_\Omega\|_F^2,
\end{aligned}$$

*where the constant $t = \sqrt{2}$.*

*Proof.* Define $M := UU^T$. By Taylor's expansion, we have

$$\begin{aligned}
&\left[\nabla^2 h(U)\right](\Delta, \Delta) \\
={}& 2\left\langle \nabla f(M_\Omega), \left(\Delta\Delta^T\right)_\Omega \right\rangle + \left[\nabla^2 f(M_\Omega; M_\Omega^*)\right]\left[(\Delta U^T + U\Delta^T)_\Omega, (\Delta U^T + U\Delta^T)_\Omega\right] \\
={}& 4\langle \nabla h(U), \Delta \rangle - 4\left\langle \nabla f(M_\Omega), (M - M^*)_\Omega \right\rangle - 2\left\langle \nabla f(M_\Omega), \left(\Delta\Delta^T\right)_\Omega \right\rangle \\
&+ \left[\nabla^2 f(M_\Omega; M_\Omega^*)\right]\left[(\Delta U^T + U\Delta^T)_\Omega, (\Delta U^T + U\Delta^T)_\Omega\right] \\
={}& 4\langle \nabla h(U), \Delta \rangle - 4\int_0^1 \left[\nabla^2 f(M_\Omega + t(M - M^*)_\Omega; M_\Omega^*)\right]\left[(M - M^*)_\Omega, (M - M^*)_\Omega\right] \, dt \\
&- 2\int_0^1 \left[\nabla^2 f(M_\Omega + t(M - M^*)_\Omega; M_\Omega^*)\right]\left[(M - M^*)_\Omega, \left(\Delta\Delta^T\right)_\Omega\right] \, dt \\
&+ \left[\nabla^2 f(M_\Omega; M_\Omega^*)\right]\left[(\Delta U^T + U\Delta^T)_\Omega, (\Delta U^T + U\Delta^T)_\Omega\right].
\end{aligned}$$

Using the $(\delta, \Omega)$-RIP$_{2r,2r}$ condition, it follows that

(17)
$$\begin{aligned}
\left[\nabla^2 h(U)\right](\Delta, \Delta) \leq{}& 4\langle \nabla h(U), \Delta \rangle - 4(1 - \delta)\|(M - M^*)_\Omega\|_F^2 - 2\left\langle (M - M^*)_\Omega, \left(\Delta\Delta^T\right)_\Omega \right\rangle \\
&+ 4\delta\|(M - M^*)_\Omega\|_F \|(\Delta\Delta^T)_\Omega\|_F + (1 + \delta)\|(\Delta U^T + U\Delta^T)_\Omega\|_F^2 \\
={}& 4\langle \nabla h(U), \Delta \rangle - (3 - 5\delta)\|(M - M^*)_\Omega\|_F^2 + (1 + \delta)\|(\Delta\Delta^T)_\Omega\|_F^2 \\
&+ 4\delta\|(M - M^*)_\Omega\|_F \|(\Delta\Delta^T)_\Omega\|_F + 2\delta\left\langle (M - M^*)_\Omega, \left(\Delta\Delta^T\right)_\Omega \right\rangle \\
\leq{}& 4\langle \nabla h(U), \Delta \rangle - (3 - 5\delta)\|(M - M^*)_\Omega\|_F^2 + (1 + \delta)\|(\Delta\Delta^T)_\Omega\|_F^2 \\
&+ 6\delta\|(M - M^*)_\Omega\|_F \|(\Delta\Delta^T)_\Omega\|_F,
\end{aligned}$$

where we have used the relation $\Delta U^T + U\Delta^T = M - M^* + \Delta\Delta^T$. Using Hölder's inequality, we have

$$2\|(M - M^*)_\Omega\|_F\|(\Delta\Delta^T)_\Omega\|_F \leq t\|(M - M^*)_\Omega\|_F^2 + t^{-1}\|(\Delta\Delta^T)_\Omega\|_F^2.$$

Substituting the above inequality into (17), we obtain

$$\left[\nabla^2 h(U)\right](\Delta, \Delta) \leq 4\langle\nabla h(U), \Delta\rangle - [3 - (5 + 3t)\delta] \cdot \|(M - M^*)_\Omega\|_F^2$$
$$+ [1 + (1 + 3t^{-1})\delta] \cdot \|(\Delta\Delta^T)_\Omega\|_F^2.$$

This is the desired result. $\qquad\square$

The following lemma is a generalization of Lemma 10 in [17].

**Lemma 4.** *Suppose that $U$ is an $\epsilon$-approximate first-order critical point of problem* (12) *for a sufficiently small $\epsilon$ and*

$$\alpha^2 = \Theta\left(\frac{\mu r\sigma_1^*}{n}\right), \quad \lambda = \Theta\left[\frac{(\sqrt{\mu} + \sqrt{n}\delta)n}{\sqrt{\mu}r}\right], \quad np \geq C\mu^2 r^3(\kappa^*)^2\log n, \quad \delta < \frac{1}{16},$$

*where $C > 0$ is a sufficiently large constant. Then, it holds with probability at least $1 - 1/\text{poly}(n)$ that*

$$\frac{1}{p}\left[-[3 - (5 + 3t)\delta] \cdot \|(M - M^*)_\Omega\|_F^2 + [1 + (1 + 3t^{-1})\delta] \cdot \|(\Delta\Delta^T)_\Omega\|_F^2\right] \leq -0.03\sigma_r^*\|\Delta\|_F^2,$$

*where the constant $t = \sqrt{2}$ and $\Delta \in \mathbb{R}^{n\times r}$ is defined in Lemma* 3.

*Proof.* By Lemma 2, we can bound the norm of each row of $U$ by

$$\max_i \|e_i^T U\|_F^2 = O\left(\frac{\mu r\sigma_1^*}{n}\right)$$

with probability at least $1 - 1/\text{poly}(n)$. Then, we split the proof into two different cases.

**Case I.** We first consider the case when $\|\Delta\|_F^2 \leq \sigma_r^*/4$. We can calculate that

$$(18) \quad \frac{1}{p}\left[-[3 - (5 + 3t)\delta] \cdot \|(M - M^*)_\Omega\|_F^2 + [1 + (1 + 3t^{-1})\delta] \cdot \|(\Delta\Delta^T)_\Omega\|_F^2\right]$$

$$= -\frac{4[3 - (5 + 3t)\delta]}{p}\left[\langle(U^*\Delta^T)_\Omega, (\Delta\Delta^T)_\Omega\rangle + \|(U^*\Delta^T)_\Omega\|_F^2\right]$$

$$+ [1 + (1 + 3t^{-1})\delta - 3 + (5 + 3t)\delta]\|(\Delta\Delta^T)_\Omega\|_F^2$$

$$\leq -\frac{9}{p}\left[\langle(U^*\Delta^T)_\Omega, (\Delta\Delta^T)_\Omega\rangle + \|(U^*\Delta^T)_\Omega\|_F^2\right]$$

$$\leq -\frac{9}{p} \cdot \|(U^*\Delta^T)_\Omega\|_F\left[\|(U^*\Delta^T)_\Omega\|_F - \|(\Delta\Delta^T)_\Omega\|_F\right],$$

where the first inequality is from the condition that $\delta < 1/16$ and $t = \sqrt{2}$. Using a similar analysis as Lemma 10 of [17] and the condition $\delta < 1/16$, it holds with probability at least $1 - 1/\text{poly}(n)$ that

$$\frac{1}{p}\|(U^*\Delta^T)_\Omega\|_F^2 \geq (1 - \nu)\sigma_r^*\|\Delta\|_F^2, \quad \frac{1}{p}\|(\Delta\Delta^T)_\Omega\|_F^2 \leq \frac{\sigma_r^*}{2}\|\Delta\|_F^2,$$

where constant $\nu > 0$ can be made sufficiently small by choosing a large enough $C$. Substituting the above bounds into inequality (18), it holds with the same probability that

$$\frac{1}{p}\left[-[3 - (5 + 3t)\delta] \cdot \|(M - M^*)_\Omega\|_F^2 + [1 + (1 + 3t^{-1})\delta] \cdot \|(\Delta\Delta^T)_\Omega\|_F^2\right]$$

$$\leq -9\sqrt{1 - \nu}\left(\sqrt{1 - \nu} - 1/\sqrt{2}\right)\sigma_r^*\|\Delta\|_F^2 < -0.03\sigma_r^*\|\Delta\|_F^2,$$

where the last inequality is by choosing a sufficiently small $\nu$.

**Case II.** Now, we consider the case when $\|\Delta\|_F^2 \geq \sigma_r^*/4$. By a similar analysis as Lemma 10 of [17], it holds with probability at least $1 - 1/\text{poly}(n)$ that

$$\frac{1}{p}\|(\Delta\Delta^T)_\Omega\|_F^2 \leq \|\Delta\Delta^T\|_F^2 + \nu\sigma_r^*\|\Delta\|_F^2,$$

$$\frac{1}{p}\|(M - M^*)_\Omega\|_F^2 \geq (1 - \nu)\|M - M^*\|_F^2 - \nu\sigma_r^*\|\Delta\|_F^2,$$

where constant $\nu > 0$ can be made sufficiently small by choosing a large enough $C$. Therefore, the condition $\delta < 1/16$ implies that with the same probability, we have

$$\frac{1}{p}\left[-[3 - (5 + 3t)\delta] \cdot \|(M - M^*)_\Omega\|_F^2 + [1 + (1 + 3t^{-1})\delta] \cdot \|(\Delta\Delta^T)_\Omega\|_F^2\right]$$

$$\leq [1 + (1 + 3t^{-1})\delta] \cdot \left(\|\Delta\Delta^T\|_F^2 + \nu\sigma_r^*\|\Delta\|_F^2\right)$$
$$\qquad - [3 - (5 + 3t)\delta] \cdot \left[(1 - \nu)\|M - M^*\|_F^2 - \nu\sigma_r^*\|\Delta\|_F^2\right]$$

$$\leq \left[2 + 2(1 + 3t^{-1})\delta - (1 - \nu)[3 - (5 + 3t)\delta]\right] \cdot \|M - M^*\|_F^2$$
$$\qquad + [1 + (1 + 3t^{-1})\delta - 3 + (5 + 3t)\delta] \cdot \nu\sigma_r^*\|\Delta\|_F^2$$

$$\leq \left[-1 + (7 + 3t + 6t^{-1})\delta + O(\nu)\right] \cdot 2(\sqrt{2} - 1)\sigma_r^*\|\Delta\|_F^2$$

$$= 2(\sqrt{2} - 1)\left[-1 + (7 + 6\sqrt{2})\delta + O(\nu)\right] \cdot \sigma_r^*\|\Delta\|_F^2 < -0.03\sigma_r^*\|\Delta\|_F^2,$$

where the second inequality is from $\|\Delta\Delta^T\|_F^2 \leq 2\|M - M^*\|_F^2$, the second last inequality is from $2(\sqrt{2} - 1)\|\Delta\|_F^2 \leq 2\|M - M^*\|_F^2$ and the last inequality is by choosing a sufficiently small $\nu$.

Combining the two cases completes the proof. $\qquad\square$

By the same proof as that of Lemma 11 of [17], we can bound the curvature of the regularizer.

**Lemma 5.** *Suppose that $U$ is an $\epsilon$-approximate first-order critical point of problem* (12) *for a sufficiently small $\epsilon$ and*

$$\alpha^2 = \Theta\left(\frac{\mu r \sigma_1^*}{n}\right), \quad \lambda \geq 0,$$

*Then, it holds with probability at least $1 - 1/\text{poly}(n)$ that*

$$\left[\nabla^2 g(U)\right](\Delta, \Delta) - 4\langle\nabla g(U), \Delta\rangle \leq 0,$$

*where $\Delta \in \mathbb{R}^{n \times r}$ is defined in Lemma* 3.

*Proof.* For each $i \in [n]$, since the regularizer is non-zero only if $\|e_i^T U\| \geq 9\alpha$, we only need to consider the index set

$$\mathcal{I} := \{i \in [n] \mid \|e_i^T U\|_F \geq 9\alpha\}.$$

We can calculate that

$$\left[\nabla^2 g(U)\right](\Delta, \Delta) - 4\langle\nabla g(U), \Delta\rangle$$

$$= 4\lambda \int_{-1}^1 \sum_{i \in \mathcal{I}} \left[\left(\|e_i^T U\|_F + \alpha y - 10\alpha\right)_+ + 8\alpha\right]^3 \frac{\|e_i^T U\|_F \|e_i^T \Delta\|_F - e_i^T U \Delta^T e_i}{\|e_i^T U\|_F^3}(1 - |y|)\,dy$$

$$+ 12\lambda \int_{-1}^1 \sum_{i \in \mathcal{I}} \left[\left(\|e_i^T U\|_F + \alpha y - 10\alpha\right)_+ + 8\alpha\right]^2 \left(\frac{e_i^T U \Delta^T e_i}{\|e_i^T U\|_F}\right)^2 (1 - |y|)\,dy$$

$$- 16\lambda \int_{-1}^1 \sum_{i \in \mathcal{I}} \left[\left(\|e_i^T U\|_F + \alpha y - 10\alpha\right)_+ + 8\alpha\right]^3 \frac{e_i^T U \Delta^T e_i}{\|e_i^T U\|_F}(1 - |y|)\,dy$$

$$= I_1 + I_2,$$

where we define

$$I_1 := 4\lambda \int_{-1}^{1} \sum_{i \in \mathcal{I}} \left[ \left( \|e_i^T U\|_F + \alpha y - 10\alpha \right)_+ + 8\alpha \right]^3$$

$$\cdot \left( \frac{\|e_i^T U\|_F \|e_i^T \Delta\|_F - e_i^T U \Delta^T e_i}{\|e_i^T U\|_F^3} - 0.4 \cdot \frac{e_i^T U \Delta^T e_i}{\|e_i^T U\|_F} \right) (1 - |y|)\, dy$$

$$I_2 := 12\lambda \int_{-1}^{1} \sum_{i \in \mathcal{I}} \left[ \left( \|e_i^T U\|_F + \alpha y - 10\alpha \right)_+ + 8\alpha \right]^2$$

$$\cdot \left( \frac{e_i^T U \Delta^T e_i}{\|e_i^T U\|_F} - 1.2 \cdot \frac{e_i^T U \Delta^T e_i}{\|e_i^T U\|_F} \right)^2 (1 - |y|)\, dy.$$

It is proved in Lemma 11 of [17] that

$$\frac{\|e_i^T U\|_F \|e_i^T \Delta\|_F - e_i^T U \Delta^T e_i}{\|e_i^T U\|_F^3} - 0.4 \cdot \frac{e_i^T U \Delta^T e_i}{\|e_i^T U\|_F} < 0,$$

which implies that

$$I_1 < 0.$$

Similarly, since we assume $\|e_i^T U\|_F \geq 9\alpha$, the second case of Lemma 11 of [17] implies that

$$\frac{e_i^T U \Delta^T e_i}{\|e_i^T U\|_F} - 1.2 \cdot \frac{e_i^T U \Delta^T e_i}{\|e_i^T U\|_F} \leq 0,$$

which leads to

$$I_2 \leq 0.$$

Hence, we get $I_1 + I_2 \leq 0$ and

$$\left[ \nabla^2 g(U) \right] (\Delta, \Delta) - 4 \langle \nabla g(U), \Delta \rangle \leq 0,$$

which is the desired result. $\hfill\square$

The next lemma establishes the bound on the curvature along $\Delta$ for an $\epsilon$-approximate first-order critical point.

**Lemma 6.** *Suppose that $U$ is an $\epsilon$-approximate first-order critical point of problem* (12) *for a sufficiently small $\epsilon$ and*

$$\alpha^2 = \Theta\left( \frac{\mu r \sigma_1^*}{n} \right), \quad \lambda = \Theta\left[ \frac{(\sqrt{\mu} + \sqrt{n}\delta)n}{\sqrt{\mu}r} \right], \quad np \geq C\mu^2 r^3 (\kappa^*)^2 \log n, \quad \delta < \frac{1}{16},$$

*Then, it holds with probability at least $1 - 1/\mathrm{poly}(n)$ that*

$$\left[ \frac{1}{p} \nabla^2 h(U) + \nabla^2 g(U) \right] (\Delta, \Delta) \leq -0.03 \sigma_r^* \|\Delta\|_F^2 + 4\epsilon \|\Delta\|_F,$$

*where $\Delta \in \mathbb{R}^{n \times r}$ is defined in Lemma 3.*

*Proof.* With probability at least $1 - 1/\mathrm{poly}(n)$, we have

$$\left[ \frac{1}{p} \nabla^2 h(U) + \nabla^2 g(U) \right] (\Delta, \Delta)$$

$$\leq 4 \left\langle \frac{1}{p} \nabla h(U) + \nabla g(U), \Delta \right\rangle + \left[ \nabla^2 r(U) \right] (\Delta, \Delta) - 4 \langle \nabla g(U), \Delta \rangle$$

$$+ \frac{1}{p} \left[ -[3 - (5 + 3t)\delta] \cdot \|(M - M^*)_\Omega\|_F^2 + [1 + (1 + 3t^{-1})\delta] \cdot \|(\Delta \Delta^T)_\Omega\|_F^2 \right]$$

$$\leq \epsilon \|\Delta\|_F - 0.03 \sigma_r^* \|\Delta\|_F^2,$$

where the first inequality is by Lemma 3 and the second inequality is by Lemmas 4 and 5. This finishes the proof of the lemma. $\hfill\square$

Using Lemmas 1-6, we are ready to prove the main theorem.

*Proof of Theorem 3.* Define $\Delta \in \mathbb{R}^{n \times r}$ in the same way as in Lemma 3. Suppose that $U$ is an $\epsilon$-approximate first-order critical point of problem (12) for a sufficiently small $\epsilon$ and

$$\operatorname{dist}(U, U^*) = \|\Delta\|_F \geq \frac{100\epsilon}{\sigma_r^*}.$$

Then, Lemma 6 implies that

$$\left[ \frac{1}{p} \nabla^2 h(U) + \nabla^2 g(U) \right] (\Delta, \Delta) \leq \epsilon \|\Delta\|_F - 0.03\sigma_r^* \|\Delta\|_F^2 \leq -\frac{200\epsilon^2}{\sigma_r^*}$$

holds with probability at least $1 - 1/\operatorname{poly}(n)$. Hence, with the same probability, the MSoG problem (12) satisfies the $(\theta, \beta, \gamma)$-strict saddle property with

$$\theta = 100\epsilon/\sigma_r^*, \quad \beta = \epsilon, \quad \gamma = 200\epsilon^2/\sigma_r^*.$$

This completes the proof. $\qquad\qquad\square$

# B Proof of Theorem 4

We split the proof into two parts. We first prove that the $(1/2, \Omega)$-RIP$_{2r,2r}$ condition holds for all non-empty $\Omega$ and then prove the existence of spurious second-order critical points.

**Proof of the $\Omega$-RIP condition.** Since the loss function $f_{3/2}$ is a quadratic function, it holds that

$$[\nabla^2 f_{3/2}(M_\Omega; M_\Omega^*)](K, K) = \frac{3}{2}\|K_\Omega\|_F^2 + K_\Omega : \mathcal{H} : K_\Omega, \quad \forall K, M \in \mathbb{R}^{n \times n}.$$

For the notational simplicity, we define

$$\tilde{K} := K_\Omega.$$

By the definition of tensor $\mathcal{H}$, we can calculate that

$$
\begin{aligned}
K_\Omega : \mathcal{H} : K_\Omega = \sum_{i \in [r]} &\Big[ -\frac{1}{2}\left(\tilde{K}_{2i-1,2i-1}^2 + \tilde{K}_{2i,2i}^2\right) + \tilde{K}_{2i-1,2i-1}\tilde{K}_{2i,2i} \\
&- \frac{1}{4}\left(\tilde{K}_{2i-1,2i}^2 + \tilde{K}_{2i,2i-1}^2 - 2\tilde{K}_{2i-1,2i}\tilde{K}_{2i,2i-1}\right)\Big] \\
= -\frac{1}{2}\sum_{i \in [r]} &\left(\tilde{K}_{2i-1,2i-1} - \tilde{K}_{2i,2i}\right)^2 - \frac{1}{4}\sum_{i \in [r]}\left(\tilde{K}_{2i-1,2i} - \tilde{K}_{2i,2i-1}\right)^2.
\end{aligned}
$$

It is straightforward to see that

(19)
$$K_\Omega : \mathcal{H} : K_\Omega \leq 0.$$

For all real numbers $a, b$, we have the inequality $(a - b)^2 \leq 2(a^2 + b^2)$. This inequality leads to

$$
\begin{aligned}
K_\Omega : \mathcal{H} : K_\Omega &\geq -\sum_{i \in [r]}\left(\tilde{K}_{2i-1,2i-1}^2 + \tilde{K}_{2i,2i}^2\right) - \frac{1}{2}\sum_{i \in [r]}\left(\tilde{K}_{2i-1,2i}^2 + \tilde{K}_{2i,2i-1}^2\right) \\
&\geq -\|\tilde{K}\|_F^2 = -\|K_\Omega\|_F^2,
\end{aligned}
$$

where the last equality is from the definition of $\tilde{K}$. Combining with inequality (19), it follows that

$$-\|K_\Omega\|_F^2 \leq K_\Omega : \mathcal{H} : K_\Omega \leq 0.$$

Hence, we have

$$\frac{1}{2}\|K_\Omega\|_F^2 \leq [\nabla^2 f_{3/2}(M_\Omega; M_\Omega^*)](K, K) \leq \frac{3}{2}\|K_\Omega\|_F^2, \quad \forall K, M \in \mathbb{R}^{n \times n},$$

which further implies the $(1/2, \Omega)$-RIP$_{2r,2r}$ condition of $f_{3/2}$.

**Existence of spurious second-order critical points.** Now, we prove the existence of a spurious second-order critical point by explicit construction. For all $i, j \in [n]$, we define

$$\omega_{i,j} := \begin{cases} 1, & \text{if } (i,j) \in \Omega \\ 0, & \text{otherwise.} \end{cases}$$

Note that we can choose $\alpha$ to be large enough so that

$$\alpha \geq 2 \max_{i \in [n]} \|e_i^T U^*\|_F = 2.$$

Otherwise, the ground truth $U^*$ is not a global minimum of problem (12) since the regularizer has a non-zero gradient at $U^*$. This is consistent with our choice of $\alpha$ in Theorem 3, because the incoherence of $M^*$ is $\mu = \sqrt{n/r}$ and $\alpha^2$ is on the order of $\Theta(1)$. We consider two different cases.

**Case I.** We first consider the case when

$$(20) \qquad p^{\frac{r(r+1)}{2}} \leq \frac{\sqrt{5}-1}{2}.$$

In this case, we show that the loss function $f_{3/2}$ has multiple global minima with probability at least $(3-\sqrt{5})/2$. By the condition (20), we can estimate the probability

$$(21) \qquad \mathbb{P}\left(\omega_{2i,2j} = 1, \ \forall i, j \in [r]\right) = p^{\frac{r(r+1)}{2}} \leq \frac{\sqrt{5}-1}{2}.$$

This is because $\omega_{2i,2j} = \omega_{2j,2i}$ for all $i, j \in [r]$ and thus, there are $r(r+1)/2$ independent Bernoulli random variables with parameter $p$. Suppose that the event in (21) does not happen (this event has probability $(3-\sqrt{5})/2$) and $\omega_{2i,2j} = 0$ for some $i, j \in [r]$. Then, we consider the matrix

$$\tilde{M}^* := M^* + \epsilon \cdot e_{2i}e_{2j}^T + \epsilon \cdot e_{2j}e_{2i}^T,$$

where $\epsilon > 0$ is sufficiently small. We can verify that $\tilde{M}^*$ is a PSD matrix and

$$\tilde{M}_\Omega^* = M_\Omega^*, \quad \text{which further leads to } f_{3/2}(\tilde{M}_\Omega^*; M_\Omega^*) = 0.$$

This implies that when the event in (21) does not happen, there exists a global minimum of $f_{3/2}$ that is different from $M^*$. Therefore, function $h(U)$ also has a global minimum $\tilde{U}^*$ such that $\tilde{U}^*(\tilde{U}^*)^T = \tilde{M}^* \neq M^*$. Note that we can choose $\epsilon$ to be small enough such that

$$\alpha \geq \max_{i \in [n]} \|e_i^T \tilde{U}^*\|_F.$$

Then, we know that the regularizer $r(U)$ is zero at $\tilde{U}^*$ and $\tilde{U}^*$ is a spurious second-order critical point of problem (12).

**Case II.** Now, we consider the case when (20) does not hold, namely,

$$p^{\frac{r(r+1)}{2}} \geq \frac{\sqrt{5}-1}{2}.$$

In this case, a similar calculation to (20) leads to

$$(22) \qquad \mathbb{P}\left(\omega_{2i,2i} = \omega_{2i-1,j} = 1, \ \forall i, j \in [r]\right) \geq p^{r(r+1)} \geq \frac{3-\sqrt{5}}{2}.$$

We focus on the case when the event in (22) holds. Define

$$U_0 := \frac{1}{\sqrt{2}}\begin{bmatrix} e_2 & e_4 & \cdots & e_{2r} \end{bmatrix}.$$

It is straightforward that

$$U_0 U_0^T = \frac{1}{2}\sum_{i \in [r]} e_{2i}e_{2i}^T \neq M^*.$$

We prove that $U_0$ is a second-order critical point of problem (12). By the construction of $U_0$, we have

$$\|e_i^T U_0\|_F \leq \frac{1}{\sqrt{2}} \leq \alpha, \quad \forall i \in [n].$$

Hence, the regularizer $r(U)$ does not contribute to the local landscape of problem (12) around point $U_0$ and we only need to prove that $U_0$ is a second-order critical point of $h(U)$.

For the first-order optimality condition, we can calculate that

$$(23) \qquad \nabla f_{3/2}\left[\left(U_0 U_0^T\right)_\Omega; M_\Omega^*\right] = \sum_{i \in [r]} \left(-\omega_{2i-1,2i-1} + \frac{\omega_{2i,2i}}{4}\right) e_{2i-1} e_{2i-1}^T$$
$$+ \sum_{i \in [r]} \frac{1}{2}\left(\omega_{2i,2i} - \omega_{2i-1,2i-1}\right) e_{2i} e_{2i}^T.$$

Therefore, the $i$-th column of the gradient of $h(U)$ at $U_0$ is

$$[\nabla h(U_0)]_i = 2\nabla f_{3/2}\left[\left(U_0 U_0^T\right)_\Omega; M_\Omega^*\right](U_0)_i = \sum_{i \in [r]} \left(\omega_{2i,2i} - \omega_{2i-1,2i-1}\right) \omega_{2i,2i} \cdot e_{2i}.$$

If $\omega_{2i-1,2i-1}$ is 0 for some $i \in [r]$, a similar construction as **Case I** shows that function $f_{3/2}$ has multiple global minima. Thus, we only need to consider the case when $\omega_{2i-1,2i-1} = 1$ for all $i \in [r]$ and under this condition, it holds that

$$[\nabla h(U_0)]_i = \sum_{i \in [r]} \left(\omega_{2i,2i} - 1\right) \omega_{2i,2i} \cdot e_{2i} = 0,$$

where the last equality is from the property $\omega_{2i,2i} \in \{0, 1\}$. This verifies the first-order optimality condition of $U_0$.

Next, we check the second-order necessary optimality condition for $U_0$. For all direction $K \in \mathbb{R}^{n \times r}$, the curvature of $h(U)$ at $U_0$ along $K$ is

$$\begin{aligned}
\left[\nabla^2 h(U_0)\right](K, K) &= 2\left\langle \nabla f_{3/2}[(U_0 U_0^T)_\Omega; M_\Omega^*], \left(KK^T\right)_\Omega \right\rangle \\
&\quad + \left[\nabla^2 f_{3/2}[(U_0 U_0^T)_\Omega; M_\Omega^*]\right]\left[\left(KU_0^T + U_0 K^T\right)_\Omega, \left(KU_0^T + U_0 K^T\right)_\Omega\right] \\
&= 2\left\langle \nabla f_{3/2}[(U_0 U_0^T)_\Omega; M_\Omega^*], \left(KK^T\right)_\Omega \right\rangle \\
&\quad + \left(KU_0^T + U_0 K^T\right)_\Omega : \left(\frac{3}{2} \cdot \mathcal{I} + \mathcal{H}\right) : \left(KU_0^T + U_0 K^T\right)_\Omega.
\end{aligned}$$

By the event in (22), equation (23) and the condition $\omega_{2i-1,2i-1} = 1$ for all $i \in [r]$, we have

$$\begin{aligned}
(24) \quad &2\left\langle \nabla f_{3/2}[(U_0 U_0^T)_\Omega; M_\Omega^*], \left(KK^T\right)_\Omega \right\rangle \\
&= \left\langle \sum_{i \in [r]} \left(-2\omega_{2i-1,2i-1} + \frac{\omega_{2i,2i}}{2}\right) e_{2i-1} e_{2i-1}^T \right. \\
&\qquad\qquad\qquad\qquad\qquad \left. + \sum_{i \in [r]} \left(\omega_{2i,2i} - \omega_{2i-1,2i-1}\right) e_{2i} e_{2i}^T, \left(KK^T\right)_\Omega \right\rangle \\
&= \sum_{i \in [r]} \left(-2\omega_{2i-1,2i-1} + \frac{\omega_{2i,2i}}{2}\right) \omega_{2i-1,2i-1} \|K_{2i-1}\|_F^2 \\
&\qquad\qquad\qquad\qquad\qquad + \sum_{i \in [r]} \left(\omega_{2i,2i} - \omega_{2i-1,2i-1}\right) \omega_{2i,2i} \|K_{2i}\|_F^2 \\
&= \left(-2 + \frac{\omega_{2i,2i}}{2}\right) \sum_{i \in [r]} \|K_{2i-1,:}\|_F^2,
\end{aligned}$$

where $K_{i,:}$ is the $i$-th row of $K$ for all $i \in [n]$. By the definition of $U_0$, we can calculate that

$$U_0 K^T = \frac{1}{\sqrt{2}} \begin{bmatrix} 0 & K_{:,1}^T & 0 & K_{:,2}^T & \cdots & K_{:,r}^T & 0 & \cdots \end{bmatrix}, \quad KU_0^T = \frac{1}{\sqrt{2}} \begin{bmatrix} 0 \\ K_{:,1} \\ 0 \\ K_{:,2} \\ \vdots \\ K_{:,r} \\ 0 \\ \vdots \end{bmatrix}.$$

Therefore, it holds that

$$(25) \quad \left(KU_0^T + U_0 K^T\right)_\Omega : \frac{3}{2} \cdot \mathcal{I} : \left(KU_0^T + U_0 K^T\right)_\Omega = \frac{3}{2} \left\| \left(KU_0^T + U_0 K^T\right)_\Omega \right\|_F^2$$

$$\geq 3 \sum_{i \in [r]} \omega_{2i,2i} K_{2i,i}^2 + \frac{3}{2} \sum_{i \in [r]} \sum_{j \in [r]} \omega_{2i-1,j} K_{2i-1,j}^2$$

and

$$(26) \quad \left(KU_0^T + U_0 K^T\right)_\Omega : \mathcal{H} : \left(KU_0^T + U_0 K^T\right)_\Omega = -\sum_{i \in [r]} \omega_{2i,2i} K_{2i,i}^2.$$

Combining the relations in (24)-(26), it follows that

$$\left[\nabla^2 h(U_0)\right](K,K) \geq \left(-2 + \frac{\omega_{2i,2i}}{2}\right) \sum_{i \in [r]} \|K_{2i-1,:}\|_F^2 + \frac{3}{2} \sum_{i \in [r]} \sum_{j \in [r]} \omega_{2i-1,j} K_{2i-1,j}^2$$

$$+ 2 \sum_{i \in [r]} \omega_{2i,2i} K_{2i,i}^2$$

$$= \sum_{i \in [r]} \sum_{j \in [r]} \left(-2 + \frac{\omega_{2i,2i}}{2} + \frac{3\omega_{2i-1,j}}{2}\right) K_{2i-1,j}^2 + 2 \sum_{i \in [r]} \omega_{2i,2i} K_{2i,i}^2.$$

Now, when the event in (22) happens, we have

$$\omega_{2i,2i} = \omega_{2i-1,j} = 1, \quad \forall i,j \in [r].$$

Therefore, we have

$$\left[\nabla^2 h(U_0)\right](K,K) \geq 2 \sum_{i \in [r]} \omega_{2i,2i} K_{2i,i}^2 \geq 0, \quad \forall K \in \mathbb{R}^{n \times r},$$

which is the second-order necessary optimality condition for $h(U)$. In summary, the point $U_0$ is a spurious second-order critical point of problem (12) with probability at least $(3 - \sqrt{5})/2$.

## C  Numerical Results on Power State Estimation Problem

In this section, we demonstrate the $\Omega$-RIP condition on the power state estimation problem and illustrate the success rate of the randomly initialized gradient descent method. Given the number of buses $N$, the power network $\mathcal{G} = (\mathcal{V}, \mathcal{E})$ is randomly generated by the Erdös-Rényi random graph model with parameter $p \in (0,1]$. The voltage at node $k$ is given by $v_k = x_k + \mathbf{i}y_k$, where $x_k$ and $y_k$ are independent standard normal random variables and $y_1 = 0$. For each line $(k,\ell) \in \mathcal{E}$, the admittance of the line $Y_{k\ell}$ is $P_{k\ell} + \mathbf{i}Q_{k\ell}$, where $P_{k\ell}$ and $Q_{k\ell}$ are uniformly randomly chosen from $[0, 0.1]$ and $[0.9, 1.0]$, respectively. We first empirically verify that the $\Omega$-RIP$_{2,2}$ condition holds for this example. We randomly generate $10,000$ rank-2 directions $K \in \mathbb{R}^{2N \times 2N}$ and check the curvature of the Hessian matrix of the loss function along direction $K$. We denote the maximal and the minimal curvature as $C$ and $c$, respectively. Then, the RIP constant can be estimated to be

$$\delta \approx \frac{C - c}{C + c}.$$

Table 1: Comparison of the estimated $\Omega$-RIP$_{2,2}$ constant with different number of buses $N$ and parameters $p$.

| **N** | 25 | 50 | 100 |
|---|---|---|---|
| $p = 0.1$ | 0.81 | 0.61 | 0.43 |
| $p = 0.25$ | 0.75 | 0.56 | 0.41 |
| $p = 0.5$ | 0.72 | 0.77 | 0.39 |
| $p = 1.0$ | 0.69 | 0.53 | 0.39 |

Table 2: Comparison of the success rate of randomly initialized gradient descent method with different number of buses $N$ and parameters $p$.

| **N** | 25 | 50 | 100 |
|---|---|---|---|
| $p = 0.1$ | 0.0 | 0.0 | 0.5 |
| $p = 0.25$ | 0.1 | 0.5 | 0.8 |
| $p = 0.5$ | 0.7 | 0.7 | 0.9 |
| $p = 1.0$ | 0.8 | 1.0 | 1.0 |

For the number of buses $N = 25, 50, 100$ and the parameter $p = 0.1, 0.25, 0.5, 1.0$, we independently generate 100 examples and estimate the RIP constant for each example. The maximum of the estimated RIP constant is summarized in Table 1. We can see that the $\Omega$-RIP$_{2,2}$ condition holds for all cases and with high probability, the constant is smaller for larger $N$ or larger $p$. Therefore, we expect that the RIP constant becomes smaller than $1/16$ for larger-scale problems and our theory applies.

Next, we apply the gradient descent algorithm with random initialization to solve the power state estimation problem. The initialization point $\mathbf{w}_0$ obeys the standard normal distribution and the step size is 0.001. We say that the algorithm successfully finds the global solution $\mathbf{z}$ if the test error $\|\mathbf{w}\mathbf{w}^T - \mathbf{z}\mathbf{z}^T\|_F$ is smaller than $10^{-3}$ within 10,000 iterations. We consider the same choices of the number of buses $N = 25, 50, 100$ and the parameter $p = 0.1, 0.25, 0.5, 1.0$. For each case, we generate 10 independent experiments and count the success rate over all experiments. We summarize the success rates in Table 2. We can see that the success rate grows with parameter $p$ and estimated RIP constant $\delta$.

## D    Relation with the Asymmetric Case

In existing literature, there exist two different formulations of the matrix completion problem: the positive semi-definite (PSD) case and the asymmetric case. In this work, we focus on the PSD case, which is formulated as problem (8) for matrix completion and problem (11) for MSoG. In the asymmetric case, the ground truth matrix $M^* \in \mathbb{R}^{m \times n}$ is still assumed to be rank-$r$ but is not required to be symmetric and PSD. Using the Burer-Monteiro factorization approach, the asymmetric MSoG problem can be formulated as

$$\min_{U \in \mathbb{R}^{m \times r}, V \in \mathbb{R}^{n \times r}} \frac{1}{p} f[(UV^T)_\Omega; M_\Omega^*] + \phi\|U^T U - V^T V\|_F^2 + g_1(U) + g_2(V),$$

where we define

$$g_1(U) := \sum_{i \in [m]} \lambda_1 \int_{-1}^{1} \left[ \left( \|e_i^T U\|_F + \alpha_1 y - 10\alpha_1 \right)_+ + 9\alpha_1 \right]^4 (1 - |y|) \ dy,$$

$$g_2(V) := \sum_{i \in [n]} \lambda_2 \int_{-1}^{1} \left[ \left( \|e_i^T V\|_F + \alpha_2 y - 10\alpha_2 \right)_+ + 9\alpha_2 \right]^4 (1 - |y|) \ dy,$$

and $\alpha_1, \alpha_2, \lambda_1, \lambda_2, \phi > 0$ are constants.

Although the PSD case is a special case of the asymmetric case, we note that there are many real-world applications of the MSoG problem that possess a PSD ground truth matrix. For example, the power state estimation problem can be formulated as a PSD MSoG problem. Therefore, a large

number of works on the matrix completion problem also focused on the PSD case; see, for example, [18, 17]. Moreover, the results of the PSD case can be directly extended to the asymmetric case. To be more concrete, suppose that the ground truth matrix can be decomposed into $M^* = U^*(V^*)^T$, where matrices $U^* \in \mathbb{R}^{m \times r}$ and $V^* \in \mathbb{R}^{n \times r}$. We define the matrices

$$W^* := \begin{bmatrix} U^* \\ V^* \end{bmatrix} \in \mathbb{R}^{(m+n) \times r}, \quad W := \begin{bmatrix} U \\ V \end{bmatrix} \in \mathbb{R}^{(m+n) \times r}.$$

Since

$$WW^T = \begin{bmatrix} UU^T & UV^T \\ VU^T & VV^T \end{bmatrix},$$

$$\|U^T U - V^T V\|_F^2 = \mathrm{tr}(UU^T \cdot UU^T) + \mathrm{tr}(VV^T \cdot VV^T) - 2\mathrm{tr}[UV^T \cdot (UV^T)^T],$$

the objective function of the asymmetric MSoG problem can be written as a function of $W$ and $W^*$. More importantly, the loss function term $f[(UV^T)_\Omega; M_\Omega^*]$ can be written as a function of $WW^T$ and $W^*(W^*)^T$, which are both rank-$r$ PSD matrices. Therefore, the asymmetric MSoG problem can be formulated as a PSD MSoG problem with variable $W$ and ground truth matrix $W^*(W^*)^T$. Then, by a similar analysis as Appendix B of [38] and Appendix B of [17], the theoretical results in Theorem 3 can be directly extended to the asymmetric case with the same bound on $\delta$ and $p$ up to a constant.