# OpenReview forum: "Geometric Analysis of Matrix Sensing over Graphs"
_NeurIPS.cc/2023/Conference — NeurIPS 2023 poster_

### Official Review · Reviewer_S3ob · 2023-07-06

**Soundness:** 3 good
**Presentation:** 3 good
**Contribution:** 3 good
**Rating:** 6
**Confidence:** 3

**Summary:**

This work is novel in a sense of studying matrix sensing over graphs - and provides the first theoretical results on the optimization landscape of the MSoG problem - and intruduce a condition to characterize the optimization complexity

**Strengths:**

Its a theoretical work and ideas are nice, and it generalize the matrix completion problem and builds the necessary theory for it such as \Omega-Rip condition. The matthematical analysis seem to be complete

**Weaknesses:**

real life applications and more experiments

** after rebuttal - i will change my score to 6- but it would be nice to show experiments / numerical results on Power State Estimation problem contained in the paper to argue the applicability better.

**Questions:**

-

**Limitations:**

-

---

> ### Author Rebuttal · Authors · 2023-08-06
>
> Real life applications and more experiments: we would like to thank the reviewer for the comment. We discussed the power state estimation problem and the numerical results the "application of MSoG problem" and the "numerical results" sections in the global response, respectively.
>
> Since our response is able to resolve all of the reviewer's concerns, we hope that the reviewer could kindly consider increasing the rating.

---

### Official Review · Reviewer_jSiy · 2023-07-06

**Soundness:** 2 fair
**Presentation:** 3 good
**Contribution:** 3 good
**Rating:** 6
**Confidence:** 4

**Summary:**

In this work, the author introduced a problem which they call matrix sensing over graphs. It is sort of a generalized version of PSD matrix completion problem. To fit in the new problem formulation, the author introduced a RIP type condition and showed that it characterizes the problem complexity. The major contribution of this work is that by properly defining a regularizer and under low incoherence condition, the author is able to show that strict saddle property holds for the problem with high probability which implies that the solution of saddle avoiding method converges to the global minimizer. By constructing a counterexample, the author showed that the bound achieved is tight up to a constant factor.

**Strengths:**

Overall, I think the introduced problem is interesting in theory:

1. The author did a good job comparing the existing results/conditions of matrix sensing problem with results they introduced. This helped the reviewer to easily understand their intuition behind their ideas;

2. Given the tightness of the bounds achieved, I think the theoretical results are also strong.

**Weaknesses:**

As a major weakness, I think the motivation of this work lacks support of applications. I understand it is a theory paper, but it is always good to have some story behind it, especially when the problem formulation is introduced as new and some of the major methods are similar to that in [38].

On the other hand, some of the claims in the paper is not so accurate. For instance, the matrix completion problem does not require $M^*$ to be PSD. In some matrix sensing problems, observations come with noise. So the problem studied in this work is not a general version of matrix completion or matrix sensing as claimed.

**Questions:**

1. For the sampling rate, I noticed that the term associated with rank is $r^3$. I am wondering is there any chance that this can be reduced to $r$. If not, what is the major challenge here.

2. Another questions is that what if $M^*$ is not PSD. Can you achieve similar results?

**Limitations:**

Please see the weakness section.

---

> ### Author Rebuttal · Authors · 2023-08-06
>
> 1. **This work lacks support of applications:** please see the "Application of MSoG problem" section in the global response.
>
> 2. **Generalization to the case when $M^*$ is not PSD:** we would like to thank the reviewer for the insightful comment. Please see our response point 1 to Reviewer do7X for the discussion on the generalization to the asymmetric (non-PSD) case.
>
> 3. **Observations come with noise:** we would like to thank the reviewer for pointing out this issue. We agree with the reviewer that in real-world applications, the observations often come with noise. However, the analysis for the noiseless case is also important in the sense that it can provide insights into the analysis for the noisy case. With a suitable assumption on the noise, it is common that the proofs of the noiseless case can be extended to the noisy case with minor modifications; see the analysis of the noisy matrix sensing problem in the following paper for a straightforward pathway to go from the noise-free case to the noisy case:
>
> 	>Ma, Z., Bi, Y., Lavaei, J., & Sojoudi, S. (2023). Geometric Analysis of Noisy Low-Rank Matrix Recovery in the Exact Parametrized and the Overparametrized Regimes. INFORMS Journal on Optimization.
>
> 	Therefore, although our work focused on the noiseless case, we expect that similar results will hold for the noisy case. We numerically verify this claim for the power state estimation problem in the global response, where the observations are generated with noise.
>
> 4. **Relation to [38]:** we would like to thank the reviewer for pointing out this related work. Although this work considers a similar topic as [38], the problem formulation of this work is significantly more general than that in [38]. More specifically, the work [38] only considered the matrix sensing problem, namely, the case when the sampling rate p is equal to 1. Our work is able to deal with the more general case when the sampling rate is (considerably) smaller than 1. As a result, we need to include the novel incoherence regularizer in the objective function and we need to use different techniques in the proofs. We will include a more detailed comparison to [38] in the revised paper.
>
> 5. **The cubic dependence of the sampling rate on $r$:** we would like to thank the reviewer for the helpful comment. The $r^3$ dependence in the sampling rate is inevitable using our current proofs. The major challenge is in the concentration inequality in the proof of Lemma 4. The sampling rate needs to be large enough to guarantee that $\|X_\Omega\|_F^2$ is close to $p\|X\|_F^2$ for certain matrix $X$. Moreover, to the best of authors' knowledge, for the matrix completion problem via the Burer-Monteiro factorization, the best dependence of the sampling rate on $r$ is $r^2$, which is derived in [13]. However, their results contain a much higher order of dependence on the condition number $\kappa^*$, and the MSoG problem is more general than the matrix completion problem and their results cannot be directly applied to the MSoG problem. We will include a more detailed discussion in the revised paper.
>
>
> Since our response is able to resolve all of the reviewer's concerns, we hope that the reviewer could kindly consider increasing the rating.

---

> ### Author Response · Authors · 2023-08-18
> **Follow-up**
>
> Dear Reviewer,
>
> Since the author-reviewer discussion period will end shortly, we would like to follow up to see if the reviewer has any further questions. We are more than happy to answer any questions the reviewer may have about our manuscript or our responses. If the reviewer's concerns have been fully addressed by our response, we hope that the reviewer can kindly consider raising the rating of our manuscript. Thank you for your time and your kind consideration!
>
> Best,
> Authors of Submission 3855

---

> > ### Comment · Reviewer_jSiy · 2023-08-22
> >
> > Dear authors,
> >
> > Thanks for your response, I will upgrade my score.

---

### Official Review · Reviewer_do7X · 2023-07-07

**Soundness:** 3 good
**Presentation:** 2 fair
**Contribution:** 2 fair
**Rating:** 5
**Confidence:** 3

**Summary:**

This paper introduces the matrix sensing over graphs (MSoG) problem, which encompasses matrix sensing and matrix completion problems. The paper proposes a non-convex regularized optimization method for this new problem and provides theoretical guarantees for its solvability using the gradient descent algorithm. The theoretical guarantees rely on the incoherence condition and a modified RIP condition introduced in the paper. The paper demonstrates that the suggested method can achieve improved sample complexity compared to existing results in the matrix completion problem.

**Strengths:**

* The paper presents a new framework that includes existing matrix sensing and matrix completion problems.
* The paper introduces a new incoherence regularizer and provides theoretical guarantees for it.

**Weaknesses:**

* The problem is restricted to cases where the true unknown matrix is positive semidefinite.
* The presentation is not very reader-friendly. Some notations are used without introduction, such as the condition number in equation (9) and tensor multiplication in the equation following line 246. Additionally, the use of examples defined in Section 3.2 but presented in Section 2 makes it difficult to follow the paper.


**Questions:**

* Why is $c$ multiplied by the threshold in line 262? It seems unnecessary and could affect the experimental result.
* In Theorem 3, it seems peculiar that the bound on $\gamma$ decreases as $\sigma^*_r$ increases. Shouldn't $\gamma$ increase as $\sigma^*_r$ increases to better avoid saddle points? Please clarify if I have misunderstood something.


**Limitations:**

Yes.

---

> ### Author Rebuttal · Authors · 2023-08-06
>
> 1. **The problem is restricted to cases where the true unknown matrix is positive semi-definite:** we would like to thank the reviewer for pointing out this limitation. In existing literature, there are two different formulations of the matrix completion problem: the positive semi-definite (PSD) case and the asymmetric case. Our work focused on the PSD case, which is formulated as problem (8) for matrix completion and problem (12) for MSoG. In the asymmetric case, the ground truth matrix $M^* \in \mathbb{R}^{m\times n} $ is still assumed to be rank-$r$ but is not required to be PSD. Using the Burer-Monteiro factorization approach, the asymmetric MSoG problem can be formulated as
> $$ \min_{U\in\mathbb{R}^{m\times r}, V\in\mathbb{R}^{n\times r}}~ \frac{1}{p} f[(UV^T)_\Omega;M^*_\Omega] + \phi \|U^TU - V^TV\|_F^2 + g_1(U) + g_2(V), $$
>
> where we define
>
> $$
> g_{1}(U) := \sum_{i\in[m]} \lambda_1 \int_{-1}^{1} \left[ \left( \|| e_{i}^{T} U \||F  + \alpha_1 y - 10\alpha_1 \right)_{+} + 9\alpha_1 \right]^4 \left(1 - |y|\right)~ dy,
> $$
>
> $$
> g_{2}(U) := \sum_{i\in[m]} \lambda_2 \int_{-1}^{1} \left[ \left( \|| e_{i}^{T} U \||F  + \alpha_2 y - 10\alpha_2 \right)_{+} + 9\alpha_2 \right]^4 \left(1 - |y|\right)~ dy,
> $$
>
> and $\alpha_1, \alpha_2, \lambda_1, \lambda_2, \phi > 0$ are constants.
>
> First, we note that there are many real-world applications of the MSoG problem that possess a PSD ground truth matrix. For example, the power state estimation problem can be formulated as a PSD MSoG problem. Therefore, a large number of works on the matrix completion problem also focused on the PSD case; see, for example, [17, 18].
>
> Additionally, although the asymmetric case is more general than the PSD case, the results of the PSD case can be directly extended to the asymmetric case. To be more concrete, suppose that the ground truth matrix can be decomposed into $M^* = U^*(V^*)^T$, where matrices $U^* \in\mathbb{R}^{m\times r}$ and $V^* \in\mathbb{R}^{n\times r}$. We define the matrices
> $$ W^* := \left(\begin{array}{cc} U^* \\ V^* \end{array}\right) \in \mathbb{R}^{(m + n) \times r},\quad W := \left(\begin{array}{cc} U \\ V \end{array}\right) \in \mathbb{R}^{(m + n) \times r}. $$
> Since
> $$
> WW^T = \left(\begin{array}{cc} UU^T & UV^T \\ VU^T & VV^T \end{array}\right),\quad \|U^TU - V^TV\|_F^2 = \mathrm{tr}(UU^T\cdot UU^T) + \mathrm{tr}(VV^T \cdot VV^T) - 2\mathrm{tr}[UV^T \cdot (UV^T)^T],
> $$
> the objective function of the asymmetric MSoG problem can be written as a function of $W$ and $W^*$. More importantly, the loss function term $f[(UV^T)_\Omega;M^*_\Omega]$ can be written as a function of $WW^T$ and $W^*(W^*)^T$, which are both rank-$r$ PSD matrices. Therefore, the asymmetric MSoG problem can be formulated as a PSD MSoG problem with variable $W$ and ground truth matrix $W^*(W^*)^T$. Then, by a similar analysis as Appendix B of [38] and Appendix B of [17], the theoretical results in Theorem 3 can be directly extended to the asymmetric case with the same bound on $\delta$ and $p$ up to a constant. We will include the rigorous analysis of the asymmetric case in the revised paper.
>
>
> 2. **The presentation is not very reader-friendly:** we would like to thank the reviewer for the constructive comment. We will follow the reviewer's comment to revise the presentation of the paper.
>
> 3. **The threshold in line 262:** we would like to thank the reviewer for pointing out this issue. Following the reviewer's suggestion, we have changed the threshold to 1e-3 and re-implemented the gradient descent algorithm. The updated results are included in the PDF file. Although the success rates are slightly different due to the randomness of the initialization, we can observe exactly the same pattern as the original experiment. We are happy to share the reproducible code after the review process.
>
> 4. **Relation between $\gamma$ and $\sigma_{r}^{*}$:** we would like to thank the reviewer for pointing out the point. We note that the small constant $\epsilon$ may also depend on $\sigma_r^*$. Therefore, it is hard to estimate the relation between $\gamma$ and $\sigma_{r}^{*}$ from Theorem 3. A similar bound is also derived in Theorem 4 of [17].
>
>
> Since our response is able to resolve all of the reviewer's concerns, we hope that the reviewer could kindly consider increasing the rating.

---

> ### Author Response · Authors · 2023-08-18
> **Follow-up**
>
> Dear Reviewer,
>
> Since the author-reviewer discussion period will end shortly, we would like to follow up to see if the reviewer has any further questions. We are more than happy to answer any questions the reviewer may have about our manuscript or our responses. If the reviewer's concerns have been fully addressed by our response, we hope that the reviewer can kindly consider raising the rating of our manuscript. Thank you for your time and your kind consideration!
>
> Best,
> Authors of Submission 3855

---

### Official Review · Reviewer_xgrK · 2023-07-07

**Soundness:** 3 good
**Presentation:** 3 good
**Contribution:** 3 good
**Rating:** 6
**Confidence:** 3

**Summary:**

This paper considers a novelly proposed problem of matrix sensing over graphs (MSoG), which is a general case of matrix completion and matrix sensing problems. It proposes a novel variant of RIP condition named Ω-RIP condition and analyze the optimization landscape of MSoG problem. With an improved regularizer of the incoherence, it proves that the strict saddle property holds for the MSoG problem with high probability under the incoherence condition and the Ω-RIP condition.

**Strengths:**

Originality: This paper proposes a novel problem formulation of MSoG, and gives the first theoretical result on the optimization landscape of this problem, which relies on its novelly proposed Ω-RIP condition and incoherence regularizer.

Quality: This paper provides rigorous theoretical analysis and strong theoretical results. The result is well supported by the numerical experiments.

Clarity: The logic of this article is rigorous, and the focus and contribution are clear.

**Weaknesses:**

1. The writing of this paper can be more clear. Taking section 1.3,1.4 as an example, it maybe more well organized to merge the motivating example and problem formulation in one section, the description of motivating example can be more tidy, and the theoretical result can be delayed to the section 2 after the presentation of  Ω-RIP condition. The nesting problem also exists in the presentation of numerical result.

2. The applications of MSoG problem can be stated more detailedly. For example, it will be more motivated to detailedly model the power state estimation as one case of MSoG problem.

3. The intuitions and high-level ideas of the formulation and analysis can be more detailed, for example, the intuition of the design of incoherence regularizer and what it contributes to the convenience of the analysis.

**Questions:**

1. What is the intuition of the design of incoherence regularizer? Whether and how it contributes to the convenience of the analysis and the improvement of sampling rate bound, compared with other regularizers?

2. Is the theoretic result general enough to applicate in some specific cases such as the landscape of power state estimation? Does the special incoherence regularizer affect the generalization?


**Limitations:**

Yes

---

> ### Author Rebuttal · Authors · 2023-08-06
>
> 1. **The writing of this paper can be more clear:** we would like to thank the reviewer for the constructive comment. We will follow the reviewer's suggestion to revise the presentation of the paper.
>
>
> 2. **Application of MSoG problem:** please see the "application of MSoG problem" section in the global response.
>
>
> 3. **The intuitions and high-level ideas of the formulation and analysis:** we would like to thank the reviewer for the helpful comment. Let $\mathcal{R}\subset \mathbb{R}^{n\times r}$ be the set of $U$ such that $\|e_i^T U\|_F = O(\alpha)$ for all $i$. The intuition behind the design of the incoherence regularizer is to ensure that every $\epsilon$-approximate first-order critical point of problem (12) must lie in set $\mathcal{R}$. More specifically, the regularizer ensures that (i) outside $\mathcal{R}$, the gradient of the regularizer is large enough such that any $\epsilon$-approximate first-order critical point of problem (12) must lie in $\mathcal{R}$, (ii) the regularizer, and thus its Hessian matrix, has a sufficiently small contribution to the objective function of problem (12) in $\mathcal{R}$ and (iii) the regularizer is twice differentiable. Please see Lemmas 1-2 for point (i) and Lemma 5 for point (ii). Therefore, using points (i) and (ii), we know that every $\epsilon$-approximate first-order critical point $U$ of problem (12) must lie in $\mathcal{R}$ and the landscape around $U$ can be approximated by that of the first term in the objective function (namely, $(1/p) \cdot h(U)$). Now, it remains show that $h(U)$ has a benign landscape in $\mathcal{R}$, which we established by a proof that is similar but considerably more complicated than existing proofs. We will include a more detailed description of high-level ideas behind the proofs and the design of the regularizer in the revised paper.
>
> 4. **Application to the power state estimation problem:** please see the "application of MSoG problem" and the "numerical results" sections in the global response for the theoretical discussion and the numerical results of the power state estimation problem, respectively.
>
> 5. **Does the special incoherence regularizer affect the generalization:** we would like to thank the reviewer for proposing the question. Since the regularizer can be evaluated in closed-form, our results can be applied to the power state estimation problem as long as the parameters $\alpha$ and $\lambda$ can be suitably chosen. In addition, we note that most existing works on matrix completion utilized the regularizer in problem (8), which has a similar form as our new regularizer and also faces the problem of choosing the parameters $\alpha$ and $\lambda$.

---

### Author Rebuttal · Authors · 2023-08-06

1. **Application of MSoG problem:** we show that the power state estimation problem [24,39] can be formulated as the MSoG problem (12). This is the most important data-analysis problem for power systems and it is solved every 5-15 minutes in practice using heuristic methods with little theoretical guarantees. Several major blackouts were attributed to the failure of solving this problem. Consider an $N$-bus power system associated with an undirected graph $G=(V, E)$, where $V=\{1,\dots,N\}$ is the set of nodes (buses) and $E \subset V\times V$ is the set of edges (lines). An edge $(k, \ell)$ belongs to $E$ if and only if there is a line connecting buses $k$ and $\ell$. The state of the power system is characterized by the voltage phasors $ v \in \mathbb{C}^N$. Our goal is to recover the voltage phasors via measurements on the voltage magnitude and the active power flow over each line. Suppose that $v = x + \mathbf{i} y$ for $x,y\in\mathbb{R}^N$. Define
$$ z := \left( \begin{array}{cc} x \\ y \end{array} \right) \in \mathbb{R}^{2N}. $$
- For each bus $k$, the voltage magnitude is defined as $|v_k|^2$, where the magnitude for a complex number $v_k = x_k + \mathbf{i}y_k$ is defined as $|v_k|^2 := x_k^2 + y_k^2$. The voltage magnitude can be written as:
$$ |v_k|^2 = z^T(e_ke_k^T + e_{N+k}e_{N+k}^T) z, $$
where $e_k$ is the $k$-th standard basis of $\mathbb{R}^N$.

- For each line $(k,\ell) \in E$, let $Y_{k\ell}\in\mathbb{C}$ be the admittance of the line. The active power flow from bus $k$ to bus $\ell$ is
$$ p_{k\ell} = \mathrm{Re}\left( v_k(v_k - v_\ell)^*Y_{k\ell}^* \right), $$
where $z^*$ is the conjugate of complex number $z$. We observe the active power flows $p_{k\ell}$ and $p_{\ell k}$ for all lines $(k,\ell)\in E$. Denote $Y_{k\ell} := P_{k\ell} + \mathbf{i} Q_{k\ell}$ for $P_{k\ell},Q_{k\ell}\in\mathbb{R}$ for all $(k,\ell) \in E$. Then, the active power flow is equal to

$$ p_{k\ell} = z^T\left[ P_{k\ell} \left( e_ke_k^T - e_ke_\ell^T + e_{N+k}e_{N+k}^T - e_{N+k}e_{N+\ell}^T \right)  + Q_{k\ell} \left( e_ke_{N+\ell}^T - e_\ell e_{N+k}^T \right) \right]z. $$

Therefore, these observations can be written as the quadratic form $z M_j z^T$ for matrices $M_1,\dots,M_m\in\mathbb{R}^{2N\times 2N}$, where $m=N+2|E|$ is the number of observations. For each sensing matrix $M_j$, only the diagonal entries and $(k,\ell),(k,\ell+N),(k+N,\ell),(k+N,\ell+N)$ entries for $(k,\ell)\in E$ may be nonzero. In addition, we fix $y_1 = z_{N + 1} = 0$ since a unitary transformation on $v=x+\mathbf{i}y$ will not change the loss function. As a result, the problem is a $(2N-1)$-dimensional rank-$1$ MSoG problem:
$$ \min_{w\in\mathbb{R}^{2N}}~ \sum_{j=1}^m \left\langle M_j, ww^T - zz^T \right\rangle^2,\quad \mathrm{s.t.}~ w_{N+1} = 0. $$
In the numerical results section, we empirically verify that the $(\delta, E)$-$RIP_{2,2}$ condition holds for this power state estimation problem. In practice, the power state estimation problem is solved via local search methods. We provide the first mathematical guarantee on the global convergence of such methods.

2. **Numerical results:** given the number of buses $N$, the power network $G=(V,E)$ is randomly generated by the Erdos-Renyi random graph model with parameter $p\in(0,1]$. The voltage at node $k$ is given by $v_k = x_k + \mathbf{i}y_k$, where $x_k$ and $y_k$ are independent standard normal random variables and $y_1 = 0$. For each line $(k,\ell) \in E$, the admittance of the line $Y_{k\ell}$ is $P_{k\ell} + \mathbf{i} Q_{k\ell}$, where $P_{k\ell}$ and $Q_{k\ell}$ is uniformly randomly chosen from $[0,0.1]$ and $[0.9, 1]$, respectively. We first empirically verify that the $(\delta, E)$-$RIP_{2,2}$ condition holds for this example. We randomly generate $10^4$ rank-$2$ directions $K\in\mathbb{R}^{2N\times 2N}$ and check the curvature of the Hessian matrix of the loss function along direction $K$. Suppose that the maximal and the minimal curvature is $C$ and $c$, respectively. Then, the RIP constant can be estimated to be $\delta \approx (C+c) / (C-c)$. For number of buses $N=25,50,100$ and parameter $p=0.1, 0.25, 0.5, 1.0$, we independently generate $100$ examples and estimate the RIP constant for each example. The maximum of the estimated RIP constant is summarized in the following table. We can see that the $(\delta, E)$-$RIP_{2,2}$ condition holds for all cases and with high probability, the constant is smaller for larger $N$ or larger $p$. Therefore, we expect the RIP constant to be smaller than $1/16$ for larger-scale problems and our theory applies.

| N    			| 25 		| 50 		| 100		|
| -------- 		| ------- 	| ------- 	| ------- 	|
| **p = 0.1**	| 0.81    	| 0.61		| 0.43		|
| **p = 0.25**	| 0.75    	| 0.56		| 0.41		|
| **p = 0.5**	| 0.72    	| 0.55		| 0.39		|
| **p = 1.0**	| 0.69    	| 0.53		| 0.39		|

Next, we apply the gradient descent algorithm with random initialization to solve the power state estimation problem. The initialization point $w_0$ obeys the standard normal distribution and the step size is $0.001$. We say the algorithm successfully finds the global solution $zz^T$ if the test error $\|ww^T - zz^T\|$ is smaller than $10^{-3}$ within $10^4$ iterations. We consider the same choices of $N$ and $p$. For each case, we generate $10$ independent experiments and count the success rate over all experiments. We summarize the success rates in the following table. We can see that the success rate grows with parameter $p$ and estimated RIP constant $\delta$.

| N    			| 25 		| 50 		| 100		|
| -------- 		| ------- 	| ------- 	| ------- 	|
| **p = 0.1**	| 0.0    	| 0.0		| 0.5		|
| **p = 0.25**	| 0.1    	| 0.5		| 0.8		|
| **p = 0.5**	| 0.7    	| 0.7		| 0.9		|
| **p = 1.0**	| 0.8    	| 1.0		| 1.0		|

Due to the time limit, we are not able to consider larger problems in the rebuttal. We will include the discussion of the power state estimation problem, the reproducible code and the numerical results for larger problems in the revised paper.

---

### Author Response · Authors · 2023-08-14
**Follow-up**

Dear reviewers and AC,

In our responses, we have provided new real-world applications and new simulation results. We believe that the new contents are important and sufficient to address the reviewers' concerns. Therefore, we hope that the reviewers could kindly check the new results and let us know if they have further questions. We will be more than happy to answer those questions. We would like to make sure that the reviewers' concerns can be fully addressed before the author discussion period ends. Thank you for your time in reviewing the submission and reading our responses!

---

> ### Comment · Area_Chair_DqAR · 2023-08-20
> **Response to Authors' Rebuttal?**
>
> Dear Reviewers,
>
> Thanks for your hard work in reviewing this paper. The authors have already submitted their responses to your earlier comments. Could you kindly take a look and indicate whether their responses have satisfactorily addressed your comments? As the author-reviewer discussion period will end on Monday, August 21, your timely feedback not only will contribute greatly to the decision process but will also be greatly appreciated by the authors and program team.
>
> Best,
> Your AC

---

### Decision · Program_Chairs · 2023-09-21

**Decision:**

Accept (poster)

**Comment:**

The paper contains good contributions to both the theory and applications of the MSoG problem. When preparing the camera-ready version, please incorporate the reviewers' comments, particularly on improving the exposition of the technical results.